# Robust Second-Order Nonconvex Optimization and Its Application to Low Rank Matrix Sensing

**Shuyao Li**
University of Wisconsin-Madison
shuyao.li@wisc.edu

**Yu Cheng**
Brown University
yu_cheng@brown.edu

**Ilias Diakonikolas**
University of Wisconsin-Madison
ilias@cs.wisc.edu

**Jelena Diakonikolas**
University of Wisconsin-Madison
jelena@cs.wisc.edu

**Rong Ge**
Duke University
rongge@cs.duke.edu

**Stephen Wright**
University of Wisconsin-Madison
swright@cs.wisc.edu

## Abstract

Finding an approximate second-order stationary point (SOSP) is a well-studied and fundamental problem in stochastic nonconvex optimization with many applications in machine learning. However, this problem is poorly understood in the presence of outliers, limiting the use of existing nonconvex algorithms in adversarial settings.

In this paper, we study the problem of finding SOSPs in the strong contamination model, where a constant fraction of datapoints are arbitrarily corrupted. We introduce a general framework for efficiently finding an approximate SOSP with *dimension-independent* accuracy guarantees, using $\widetilde{O}(D^2/\epsilon)$ samples where $D$ is the ambient dimension and $\epsilon$ is the fraction of corrupted datapoints.

As a concrete application of our framework, we apply it to the problem of low rank matrix sensing, developing efficient and provably robust algorithms that can tolerate corruptions in both the sensing matrices and the measurements. In addition, we establish a Statistical Query lower bound providing evidence that the quadratic dependence on $D$ in the sample complexity is necessary for computationally efficient algorithms.

## 1 Introduction

Learning in the presence of corrupted data is a significant challenge in machine learning (ML) with many applications, including ML security [Bar+10; BNL12; SKL17; Dia+19] and exploratory data analysis of real datasets, e.g., in biological settings [Ros+02; Pas+10; Li+08; Dia+17]. The goal in such scenarios is to design efficient learning algorithms that can tolerate a small constant fraction of outliers and achieve error guarantees independent of the dimensionality of the data. Early work in robust statistics [Ham+86; HR09] gave sample-efficient robust estimators for various tasks (e.g., the Tukey median [Tuk75] for robust mean estimation), alas with runtimes exponential in the dimension. A recent line of work in computer science, starting with [Dia+16; LRV16], developed the first computationally efficient robust algorithms for several fundamental high-dimensional tasks. Since these early works, there has been significant progress in algorithmic aspects of robust high-dimensional statistics (see [DK19] and [DK23] for a comprehensive overview).

37th Conference on Neural Information Processing Systems (NeurIPS 2023).

In this paper, we study the general problem of smooth (with Lipschitz gradient and Hessian) stochastic nonconvex optimization $\min_x \bar{f}(x)$ in the outlier-robust setting, where $\bar{f}(x) := \mathbb{E}_{A \sim \mathcal{G}} f(x, A)$ and $\mathcal{G}$ is a possibly unknown distribution of the random parameter $A$. We will focus on the following standard adversarial contamination model (see, e.g., [Dia+16]).

**Definition 1.1** (Strong Contamination Model). Given a parameter $0 < \epsilon < 1/2$ and an inlier distribution $\mathcal{G}$, an algorithm receives samples from $\mathcal{G}$ with $\epsilon$-contamination as follows: The algorithm first specifies the number of samples $n$, and $n$ samples are drawn independently from $\mathcal{G}$. An adversary is then allowed to inspect these samples, and replace an $\epsilon$-fraction of the samples with arbitrary points. This modified set of $n$ points is said to be $\epsilon$-corrupted, which is then given to the algorithm.

The stochastic optimization problem we consider is computationally intractable in full generality — even without corruption — if the goal is to obtain globally optimal solutions. At a high level, an achievable goal is to design sample and computationally efficient robust algorithms for finding *locally* optimal solutions. Prior work [Pra+20; Dia+19] studied outlier-robust stochastic optimization and obtained efficient algorithms for finding approximate *first-order* stationary points. While first-order guarantees suffice for convex problems, it is known that in many tractable non-convex problems, first-order stationary points may be bad solutions, but all *second-order* stationary points (SOSPs) are globally optimal. This motivates us to study the following questions:

> *Can we develop a general framework for finding **second-order** stationary points in outlier-robust stochastic optimization?*
>
> *Can we obtain sample and computationally efficient algorithms for outlier-robust versions of tractable nonconvex problems using this framework?*

In this work, we answer both questions affirmatively. We introduce a framework for efficiently finding an approximate SOSP when $\epsilon$-fraction of the functions are corrupted and then use our framework to solve the problem of outlier-robust low rank matrix sensing.

In addition to the gradient being zero, a SOSP requires the Hessian matrix to not have negative eigenvalues. The second-order optimality condition is important because it rules out suboptimal solutions such as strict saddle points. It is known that all SOSPs are globally optimal in nonconvex formulations of many important machine learning problems, such as matrix completion [GLM16], matrix sensing [BNS16], phase retrieval [SQW16], phase synchronization [BBV16], dictionary learning [SQW17], and tensor decomposition [Ge+15] (see also [WM22, Chapter 7]). However, the properties of SOSPs are highly sensitive to perturbation in the input data. For example, it is possible to create spurious SOSPs for nonconvex formulations of low rank matrix recovery problems, even for a semi-random adversary that can add additional sensing matrices but cannot corrupt the measurements in matrix sensing [GC23] or an adversary who can only reveal more entries of the ground-truth matrix in matrix completion [CG18]. Those spurious SOSPs correspond to highly suboptimal solutions.

Finding SOSPs in stochastic nonconvex optimization problems in the presence of arbitrary outliers was largely unaddressed prior to our work. Prior works [Pra+20; Dia+19] obtained efficient and robust algorithms for finding *first-order* stationary points with dimension-independent accuracy guarantees. These works relied on the following simple idea: Under certain smoothness assumptions, projected gradient descent with an *approximate* gradient oracle efficiently converges to an *approximate* first-order stationary point. Moreover, in the outlier-robust setting, approximating the gradient at a specific point amounts to a robust mean estimation problem (for the underlying distribution of the gradients), which can be solved by leveraging existing algorithms for robust mean estimation.

Our work is the first to find approximate SOSPs with dimension-independent errors in outlier-robust settings. Note that in standard non-robust settings, approximate SOSPs can be computed using first-order methods such as perturbed gradient descent [Jin+17; Jin+21]. This strategy might seem extendable to outlier-robust settings through perturbed approximate gradient descent, utilizing robust mean estimation algorithms to approximate gradients. The approach in [Yin+19] follows this idea, but unfortunately their second-order guarantees scale polynomially with dimension, even under very strong distributional assumptions (e.g., subgaussianity). Our lower bound result provides evidence that approximating SOSPs with dimension-independent error is as hard as approximating *full* Hessian, suggesting that solely approximating the gradients is not sufficient. On a different note, [IPL23] recently employed robust estimators for both gradient and Hessian in solving certain convex stochastic

optimization problems, which has a different focus than ours and does not provide SOSPs with the guarantees that we achieve.

## 1.1 Our Results and Contributions

The notation we use in this section is defined in Section 2. To state our results, we first formally define our generic nonconvex optimization problem. Suppose there is a true distribution over functions $f : \mathbb{R}^D \times \mathcal{A} \to \mathbb{R}$, where $f(x, A)$ takes an argument $x \in \mathbb{R}^D$ and is parameterized by a random variable $A \in \mathcal{A}$ drawn from a distribution $\mathcal{G}$. Our goal is to find an $(\epsilon_g, \epsilon_H)$-approximate SOSP of the function $\bar{f}(x) := \mathbb{E}_{A \sim \mathcal{G}} f(x, A)$.

**Definition 1.2** ($\epsilon$-Corrupted Stochastic Optimization). The algorithm has access to $n$ functions $(f_i)_{i=1}^n$ generated as follows. First $n$ random variables $(A_i)_{i=1}^n$ are drawn independently from $\mathcal{G}$. Then an adversary arbitrarily corrupts an $\epsilon$ fraction of the $A_i$'s. Finally, the $\epsilon$-corrupted version of $f_i(\cdot) = f(\cdot, A_i)$ is sent to the algorithm as input. The task is to find an approximate SOSP of the ground-truth average function $\bar{f}(\cdot) := \mathbb{E}_{A \sim \mathcal{G}} f(\cdot, A)$.

**Definition 1.3** (Approximate SOSPs). A point $x$ is an $(\epsilon_g, \epsilon_H)$-approximate second-order stationary point (SOSP) of $\bar{f}$ if $\left\|\nabla \bar{f}(x)\right\| \leq \epsilon_g$ and $\lambda_{\min}\left(\nabla^2 \bar{f}(x)\right) \geq -\epsilon_H$.

We make the following additional assumptions on $f$ and $\mathcal{G}$.

**Assumption 1.4.** *There exists a bounded region $\mathcal{B}$ such that the following conditions hold:*

(i) *There exists a lower bound $f^* > -\infty$ such that for all $x \in \mathcal{B}$, $f(x, A) \geq f^*$ with probability 1.*

(ii) *There exist parameters $L_{D_g}$, $L_{D_H}$, $B_{D_g}$, and $B_{D_H}$ such that, with high probability over the randomness in $A \sim \mathcal{G}$, letting $g(x) = f(x, A)$, we have that $g(x)$ is $L_{D_g}$-gradient Lipschitz and $L_{D_H}$-Hessian Lipschitz over $\mathcal{B}$, and $\|\nabla g(x)\| \leq B_{D_g}$ and $\left\|\nabla^2 g(x)\right\|_F \leq B_{D_H}$ for all $x \in \mathcal{B}$.*

(iii) *There exist parameters $\sigma_g, \sigma_H > 0$ such that for all $x \in \mathcal{B}$,*

$$\left\|\mathrm{Cov}_{A \sim \mathcal{G}}(\nabla f(x, A))\right\|_{\mathrm{op}} \leq \sigma_g^2 \text{ and } \left\|\mathrm{Cov}_{A \sim \mathcal{G}}(\mathrm{vec}(\nabla^2 f(x, A)))\right\|_{\mathrm{op}} \leq \sigma_H^2.$$

Note that the radius of $\mathcal{B}$ and the parameters $L_{D_g}$, $L_{D_H}$, $B_{D_g}$, $B_{D_H}$ are all allowed to depend polynomially on $D$ and $\epsilon$ (but not on $x$ and $A$).

Our main algorithmic result for $\epsilon$-corrupted stochastic optimization is summarized in the following theorem. A formal version of this theorem is stated as Theorem 3.1 in Section 3.

**Theorem 1.5** (Finding an Outlier-Robust SOSP, informal). *Suppose $f$ satisfies Assumption 1.4 in a region $\mathcal{B}$ with parameters $\sigma_g$ and $\sigma_H$. Given an arbitrary initial point $x_0 \in \mathcal{B}$ and an $\epsilon$-corrupted set of $n = \widetilde{\Omega}(D^2/\epsilon)$ functions where $D$ is the ambient dimension, there exists a polynomial-time algorithm that with high probability outputs an $(O(\sigma_g\sqrt{\epsilon}), O(\sigma_H\sqrt{\epsilon}))$-approximate SOSP of $\bar{f}$, provided that all iterates of the algorithm stay inside $\mathcal{B}$.*

Although the bounded iterate condition in Theorem 1.5 appears restrictive, this assumption holds if the objective function satisfies a "dissipativity" property, which is a fairly general phenomenon [Hal10]. Moreover, adding an $\ell_2$-regularization term enables any Lipschitz function to satisfy the dissipativity property [RRT17, Section 4]. As an illustrating example, a simple problem-specific analysis shows that this bounded iterate condition holds for outlier-robust matrix sensing by exploiting the fact that the matrix sensing objective satisfies the dissipativity property.

In this paper, we consider the problem of outlier-robust symmetric low rank matrix sensing, which we formally define below. We focus on the setting with Gaussian design.

**Definition 1.6** (Outlier-Robust Matrix Sensing). There is an unknown rank-$r$ ground-truth matrix $M^* \in \mathbb{R}^{d \times d}$ that can be factored into $U^* U^{*\top}$ where $U^* \in \mathbb{R}^{d \times r}$. The (clean) sensing matrices $\{A_i\}_{i \in [n]}$ have i.i.d. standard Gaussian entries. The (clean) measurements $y_i$ are obtained as $y_i = \langle A_i, M^* \rangle + \zeta_i$, where the noise $\zeta_i \sim \mathcal{N}(0, \sigma^2)$ is independent from all other randomness. We denote the (clean) data generation process by $(A_i, y_i) \sim \mathcal{G}_\sigma$. When $\sigma = 0$, we have $\zeta_i = 0$ and we write $\mathcal{G} := \mathcal{G}_0$ for this noiseless (measurement) setting. In outlier robust matrix sensing, an adversary can arbitrarily change any $\epsilon$-fraction of the sensing matrices and the corresponding measurements. This corrupted set of $(A_i, y_i)$'s is then given to the algorithm as input, where the goal is to recover $M^*$.

We highlight that in our setting, both the sensing matrices $A_i \in \mathbb{R}^{d \times d}$ and the measurements $y_i \in \mathbb{R}$ can be corrupted, presenting a substantially more challenging problem compared to prior works (e.g., [Li+20b; Li+20a]) that only allow corruption in $y_i$.

Let $\sigma_1^\star$ and $\sigma_r^\star$ denote the largest and the smallest nonzero singular value of $M^*$ respectively. We assume $\sigma_r^\star$ and the rank $r$ are given to the algorithm, and we assume that the algorithm knows a multiplicative upper bound $\Gamma$ of $\sigma_1^\star$ such that $\Gamma \geq 36\sigma_1^\star$ (a standard assumption in matrix sensing even for non-robust settings [GJZ17; Jin+17]). Let $\kappa = \Gamma/\sigma_r^\star$.

Our main algorithmic result for the low rank matrix sensing problem is summarized in the following theorem. For a more detailed statement, see Theorems 3.2 and 3.3 in Section 3.

**Theorem 1.7** (Our Algorithm for Outlier-Robust Matrix Sensing). *Let $M^* \in \mathbb{R}^{d \times d}$ be the rank $r$ ground-truth matrix with smallest nonzero singular value $\sigma_r^\star$. Let $\Gamma \geq 36 \|M^*\|_{\mathrm{op}}$ and let $\kappa = \Gamma/\sigma_r^\star$. There exists an algorithm for outlier-robust matrix sensing, where an $\epsilon = O(1/(\kappa^3 r^3))$ fraction of samples from $\mathcal{G}_\sigma$ as in Definition 1.6 gets arbitrarily corrupted, that can output a rank-$r$ matrix $\widehat{M}$ such that $\|\widehat{M} - M^*\|_F \leq \iota$ with probability at least $1 - \xi$, where $\iota > 0$ is the error parameter:*

*1) If $\sigma \geq r\Gamma$, then $\iota = O(\sigma\sqrt{\epsilon})$;*

*2) If $\sigma \leq r\Gamma$, then $\iota = O(\kappa\sigma\sqrt{\epsilon})$;*

*3) If $\sigma = 0$ (noiseless), then $\iota$ can be made arbitrarily small, achieving exact recovery.*

*The algorithm uses $n = \widetilde{O}\left(\frac{d^2 r^2 + dr \log(\Gamma/\xi)}{\epsilon}\right)$ samples and runs in time $\mathrm{poly}(n, \kappa, \log(\sigma_r^\star/\iota))$.*

Finally, we complement our algorithmic results for outlier-robust matrix sensing with a Statistical Query (SQ) lower bound, which provides strong evidence that quadratic dependence on $d$ in the sample complexity is unavoidable for efficient algorithms. A detailed statement of this result is provided in Section 4.

## 1.2 Our Techniques

**Outlier-robust nonconvex optimization.** To obtain our algorithmic result in the general nonconvex setting, we leverage existing results on robust mean estimation [DKP20], which we use as a black box to robustly estimate the gradient and the (vectorized) Hessian. We use these robust estimates as a subroutine in a randomized nonconvex optimization algorithm (described in Appendix A.2), which can tolerate inexactness in both the gradient and the Hessian. With high probability, this algorithm outputs an $(\epsilon_g, \epsilon_H)$-approximate SOSP, where $\epsilon_g$ and $\epsilon_H$ depend on the inexactness of the gradient and Hessian oracles.

We remark that robust estimation of the Hessian is crucial to obtaining our *dimension-independent* approximation error result and is what causes $D^2$ dependence in the sample complexity (which is unavoidable for SQ algorithms as discussed below). Notably, the only prior work on approximating SOSPs in the outlier-robust setting [Yin+19] used robust mean estimation only on the gradients and had sample complexity scaling linearly with $D$; however, they can only output an order $(\sqrt{\epsilon}, (\epsilon D)^{1/5})$-SOSP, which is uninformative for many problems of interest, including the matrix sensing problem considered in this paper, due to the dimensional dependence in the approximation error.

**Application to low rank matrix sensing.** Our main contribution on the algorithmic side is showing that our outlier-robust nonconvex optimization framework can be applied to solve outlier-robust matrix sensing with *dimension-independent* approximation error, even achieving exact recovery when the measurements are noiseless. We obtain this result using the following geometric insights about the problem: We show that the norm of the covariance of the gradient and the Hessian can both be upper bounded by the sum of $\sigma^2$ and a function of the distance to the closest optimal solution. We further prove that all iterates stay inside a nice region using a "dissipativity" property [Hal10]), which says that the iterate aligns with the direction of the gradient when the iterate's norm is large. This allows us to invoke Theorem 1.5 to obtain an approximate SOSP of the ground-truth objective function.

We show that this approximate SOSP must be close to a global optimal solution. Additionally, we establish a local regularity condition in a small region around globally optimal solutions (which is

similar to strong convexity but holds only locally). This local regularity condition bounds below a measure of stationarity, which allows us to prove that gradient descent-type updates contract the distance to the closest global optimum under appropriate stepsize. We leverage this local regularity condition to prove that the iterates of the algorithm stay near a global optimum, so that the regularity condition continues to hold, and moreover, the distance between the current solution and the closest global optimum contracts, as long as it is larger than a function of $\sigma$. Consequently, the distance-dependent component of the gradient and Hessian covariance bound contracts as well, which allows us to obtain more accurate gradient and Hessian estimates. While such a statement may seem evident to readers familiar with linear convergence arguments, we note that proving it is quite challenging, due to the circular dependence between the distance from the current solution to global optima, the inexactness in the gradient and Hessian estimates, and the progress made by our algorithm.

The described distance-contracting argument allows us to control the covariance of the gradient and Hessian, which we utilize to recover $M^*$ exactly when $\sigma = 0$, and recover $M^*$ with error roughly $O(\sigma\sqrt{\epsilon})$ when $0 \neq \sigma \leq r\Gamma$. We note that the $\sigma\sqrt{\epsilon}$ appears unavoidable in the $\sigma \neq 0$ case, due to known limits of robust mean estimation algorithms [DKS17].

**SQ lower bound.**  We exhibit a hard instance of low rank matrix sensing problem to show that quadratic dependence on the dimension in sample complexity is unavoidable for computationally efficient algorithms. Our SQ lower bound proceeds by constructing a family of distributions, corresponding to corruptions of low rank matrix sensing, that are nearly uncorrelated in a well-defined technical sense [Fel+17]. To achieve this, we follow the framework of [DKS17] which considered a family of distributions that are rotations of a carefully constructed one-dimensional distribution. The proof builds on [DKS19; Dia+21], using a new univariate moment-matching construction which yields a family of corrupted conditional distributions. These induce a family of joint distributions that are SQ-hard to learn.

### 1.3  Roadmap

Section 2 defines the necessary notation and discusses relevant building blocks of our algorithm and analysis. Section 3 introduces our framework for finding SOSPs in the outlier-robust setting. Section 3.1 presents how to extend and apply our framework to solve outlier-robust low rank matrix sensing. Section 4 proves that our sample complexity has optimal dimensional dependence for SQ algorithms. Most proofs are deferred to the supplementary material due to space limitations.

## 2  Preliminaries

For an integer $n$, we use $[n]$ to denote the ordered set $\{1, 2, \ldots, n\}$. We use $[a_i]_{i \in \mathcal{I}}$ to denote the matrix whose columns are vectors $a_i$, where $\mathcal{I}$ is an ordered set. We use $\mathbb{1}_E(x)$ to denote the indicator function that is equal to 1 if $x \in E$ and 0 otherwise. For two functions $f$ and $g$, we say $f = \widetilde{O}(g)$ if $f = O(g \log^k(g))$ for some constant $k$, and we similarly define $\widetilde{\Omega}$.

For vectors $x$ and $y$, we let $\langle x, y \rangle$ denote the inner product $x^\top y$ and $\|x\|$ denote the $\ell_2$ norm of $x$. For $d \in \mathbb{Z}_+$, we use $I_d$ to denote the identity matrix of size $d \times d$. For matrices $A$ and $B$, we use $\|A\|_{\mathrm{op}}$ and $\|A\|_F$ to denote the spectral norm and Frobenius norm of $A$ respectively. We use $\lambda_{\max}(A)$ and $\lambda_{\min}(A)$ to denote the maximum and minimum eigenvalue of $A$ respectively. We use $\mathrm{tr}(A)$ to denote the trace of a matrix $A$. We use $\langle A, B \rangle = \mathrm{tr}(A^\top B)$ to denote the entry-wise inner product of two matrices of the same dimension. We use $\mathrm{vec}(A) = [a_1^\top, a_2^\top, \ldots, a_d^\top]^\top$ to denote the canonical flattening of $A$ into a vector, where $a_1, a_2, \ldots, a_d$ are columns of $A$.

**Definition 2.1** (Lipschitz Continuity). Let $\mathcal{X}$ and $\mathcal{Y}$ be normed vector spaces. A function $h : \mathcal{X} \to \mathcal{Y}$ is $\ell$-Lipschitz if $\|h(x_1) - h(x_2)\|_{\mathcal{Y}} \leq \ell \|x_1 - x_2\|_{\mathcal{X}}$, $\forall x_1, x_2$.

In this paper, when $\mathcal{Y}$ is a space of matrices, we take $\|\cdot\|_{\mathcal{Y}}$ to be the spectral norm $\|\cdot\|_{\mathrm{op}}$. When $\mathcal{X}$ is a space of matrices, we take $\|\cdot\|_{\mathcal{X}}$ to be the Frobenius norm $\|\cdot\|_F$; this essentially views the function $h$ as operating on the vectorized matrices endowed with the usual $\ell_2$ norm. When $\mathcal{X}$ or $\mathcal{Y}$ is the Euclidean space, we take the corresponding norm to be the $\ell_2$ norm.

**A Randomized Algorithm with Inexact Gradients and Hessians.**  We now discuss how to solve the unconstrained nonconvex optimization problem $\min_{x \in \mathbb{R}^D} f(x)$, where $f(\cdot)$ is a smooth function

with Lipschitz gradients and Lipschitz Hessians. The goal of this section is to find an approximate SOSP as defined in Definition 1.3.

**Proposition 2.2** ([LW23]). *Suppose a function $f$ is bounded below by $f^* > -\infty$, has $L_g$-Lipschitz gradient and $L_H$-Lipschitz Hessian, and its inexact gradient and Hessian computations $\widetilde{g}_t$ and $\widetilde{H}_t$ satisfy $\|\widetilde{g}_t - \nabla f(x_t)\| \leq \frac{1}{3}\epsilon_g$ and $\|\widetilde{H}_t - \nabla^2 f(x_t)\|_{\mathrm{op}} \leq \frac{2}{9}\epsilon_H$. Then there exists an algorithm (Algorithm A.1) with the following guarantees:*

1. *(Correctness) If Algorithm A.1 terminates and outputs $x_n$, then $x_n$ is a $(\frac{4}{3}\epsilon_g, \frac{4}{3}\epsilon_H)$-approximate SOSP.*

2. *(Runtime) Algorithm A.1 terminates with probability 1. Let $C_\epsilon := \min\left(\frac{\epsilon_g^2}{6L_g}, \frac{2\epsilon_H^3}{9L_H^2}\right)$. With probability at least $1 - \delta$, Algorithm A.1 terminates after $k$ iterations for*

$$k = O\Big(\frac{f(x_0) - f^*}{C_\epsilon} + \frac{L_H^2 L_g^2 \epsilon_g^2}{\epsilon_H^6}\log\Big(\frac{1}{\delta}\Big)\Big). \tag{1}$$

The constants $1/3$ and $2/9$ are chosen for ease of presentation. For all constructions of Hessian oracles in this paper, we take the straightforward relaxation $\|\widetilde{H}_t - \nabla^2 f(x_t)\|_{\mathrm{op}} \leq \|\widetilde{H}_t - \nabla^2 f(x_t)\|_F$ and upper bound Hessian inexactness using Frobenius norm. Proof of a simplified version of Proposition 2.2 with a weaker high probability bound that is sufficient for our purposes is provided in Appendix A.2 for completeness.

**Robust Mean Estimation.** Recent algorithms in robust mean estimation give dimension-independent error in the presence of outliers under strong contamination model.

We use the following results, see, e.g., [DKP20], where the upper bound $\sigma$ on the spectral norm of covariance matrix is unknown to the algorithm.

**Proposition 2.3** (Robust Mean Estimation). *Fix any $0 < \xi < 1$. Let $S$ be a multiset of $n = O((k \log k + \log(1/\xi))/\epsilon)$ i.i.d. samples from a distribution on $\mathbb{R}^k$ with mean $\mu_S$ and covariance $\Sigma$. Let $T \subset \mathbb{R}^k$ be an $\epsilon$-corrupted version of $S$ as in Definition 1.1. There exists an algorithm (Algorithm A.2) such that, with probability at least $1 - \xi$, on input $\epsilon$ and $T$ (but not $\|\Sigma\|_{\mathrm{op}}$) returns a vector $\widehat{\mu}$ in polynomial time so that $\|\mu_S - \widehat{\mu}\| = O(\sqrt{\|\Sigma\|_{\mathrm{op}}\,\epsilon})$.*

Algorithm A.2 is given in Appendix A.3. Proposition 2.3 states that for Algorithm A.2 to succeed with high probability, $\widetilde{O}(k/\epsilon)$ i.i.d. samples need to be drawn from a $k$-dimensional distribution of bounded covariance. State of the art algorithms for robust mean estimation can be implemented in near-linear time, requiring only a logarithmic number of passes on the data, see, e.g, [Dia+22; CDG19; DHL19]. Any of these faster algorithms could be used for our purposes.

With the above results, the remaining technical component for applying the robust estimation subroutine (Algorithm A.2) in this paper is handling the dependence across iterations.

Because we will run RobustMeanEstimation in each iteration of our optimization algorithm, the gradients $\{\nabla f_i(x_t)\}_{i=1}^n$ and Hessians $\{\nabla^2 f_i(x_t)\}_{i=1}^n$ can no longer be considered as independently drawn from a distribution after the first iteration. Although they are i.i.d. for fixed $x$, the dependence on previous iterations through $x$ will break the independence assumption. Therefore, we will need a union bound over all $x_t$ to handle dependence across different iterations $t$. We deal with this technicality in Appendix B.

## 3  General Robust Nonconvex Optimization

In this section, we establish a general result that uses Algorithm A.1 to obtain approximate SOSPs in the presence of outliers under strong contamination. The inexact gradient and inexact Hessian oracles are constructed with the robust mean estimation subroutine (Algorithm A.2). We consider stochastic optimization tasks in Definition 1.2 satisfying Assumption 1.4. We construct the inexact gradient and Hessian oracle required by Algorithm A.1 as follows:

$$\widetilde{g}_t \leftarrow \textbf{RobustMeanEstimation}(\{\nabla f_i(x_t)\}_{i=1}^n, 4\epsilon)$$

$$\widetilde{H}_t \leftarrow \textbf{RobustMeanEstimation}(\{\nabla^2 f_i(x_t)\}_{i=1}^n, 4\epsilon)$$

Then we have the following guarantee:

**Theorem 3.1.** *Suppose we are given $\epsilon$-corrupted set of functions $\{f_i\}_{i=1}^n$ for sample size $n$, generated according to Definition 1.2. Suppose Assumption 1.4 holds in a bounded region $\mathcal{B} \subset \mathbb{R}^D$ of diameter $\gamma$ with gradient and Hessian covariance bound $\sigma_g$ and $\sigma_H$ respectively, and we have an arbitrary initialization $x_0 \in \mathcal{B}$. Algorithm A.1 initialized at $x_0$ outputs an $(\epsilon_g, \epsilon_H)$-approximate SOSP for a sufficiently large sample with probability at least $1 - \xi$ if the following conditions hold:*

*(I) All iterates $x_t$ in Algorithm A.1 stay inside the bounded region $\mathcal{B}$.*

*(II) For an absolute constant $c > 0$, it holds that $\sigma_g \sqrt{\epsilon} \leq c\epsilon_g$ and $\sigma_H \sqrt{\epsilon} \leq c\epsilon_H$.*

*The algorithm uses $n = \widetilde{O}(D^2/\epsilon)$ samples, where $\widetilde{O}(\cdot)$ hides logarithmic dependence on $D, \epsilon, L_{D_g}, L_{D_H}, B_{D_g}, B_{D_H}, \gamma/\sigma_H, \gamma/\sigma_g$, and $1/\xi$. The algorithm runs in time polynomial in the above parameters.*

Note that we are able to obtain dimension-independent errors $\epsilon_g$ and $\epsilon_H$, provided that $\sigma_g$ and $\sigma_H$ are dimension-independent.

### 3.1 Low Rank Matrix Sensing Problems

In this section, we study the problem of outlier-robust low rank matrix sensing as formally defined in Definition 1.6. We first apply the above framework to obtain an approximate SOSP in Section 3.1.2. Then we make use of the approximate SOSP to obtain a solution that is close to the ground-truth matrix $M^*$ in Section 3.1.3; this demonstrates the usefulness of approximate SOSPs.

#### 3.1.1 Main results for Robust Low Rank Matrix Sensing

The following are the main results that we obtain in this section:

**Theorem 3.2** (Main Theorem Under Noiseless Measurements)**.** *Consider the noiseless setting as in Theorem 1.7 with $\sigma = 0$. For some sample size $n = \widetilde{O}((d^2r^2 + dr\log(\Gamma/\xi))/\epsilon)$ and with probability at least $1 - \xi$, there exists an algorithm that outputs a solution that is $\iota$-close to $M^*$ in Frobenius norm in $O(r^2\kappa^3 \log(1/\xi) + \kappa \log(\sigma_r^\star/\iota))$ calls to the robust mean estimation subroutine (Algorithm A.2).*

This result achieves exact recovery of $M^*$, despite the strong contamination of samples. Each iteration involves a subroutine call to robust mean estimation. Algorithm A.2 presented here is one simple example of robust mean estimation; there are refinements [Dia+22; DHL19] that run in nearly linear time, so the total computation utilizing those more efficient algorithms indeed requires $\widetilde{O}\left(r^2\kappa^3\right)$ passes of data (computed gradients and Hessians).

**Theorem 3.3** (Main Theorem Under Noisy Measurements)**.** *Consider the same setting as in Theorem 1.7 with $\sigma \neq 0$. There exists a sample size $n = \widetilde{O}((d^2r^2 + dr\log(\Gamma/\xi))/\epsilon)$ such that*

- *if $\sigma \leq r\Gamma$, then with probability at least $1 - \xi$, there exists an algorithm that outputs a solution $\widehat{M}$ in $\widetilde{O}(r^2\kappa^3)$ calls to robust mean estimation routine A.2, with error $\|\widehat{M} - M^*\|_F = O(\kappa\sigma\sqrt{\epsilon})$;*

- *if $\sigma \geq r\Gamma$, then with probability at least $1 - \xi$, there exists a (different) algorithm that outputs a solution $\widehat{M}$ in one call to robust mean estimation routine A.2, with error $\|\widehat{M} - M^*\|_F = O(\sigma\sqrt{\epsilon})$.*

We prove Theorem 3.3 in Appendix C.4, and instead focus on the noiseless measurements with $\sigma = 0$ when we develop our algorithms in this section; the two share many common techniques. In the remaining part of Section 3.1, we use $\mathcal{G}_0$ in Definition 1.6 for the data generation process.

We now describe how we obtain the solution via nonconvex optimization. Consider the following objective function for (uncorrupted) matrix sensing:

$$\min_{\substack{M \in \mathbb{R}^{d \times d} \\ \text{rank}(M)=r}} \frac{1}{2} \mathop{\mathbb{E}}_{(A_i, y_i) \sim \mathcal{G}_0} (\langle M, A_i \rangle - y_i)^2. \tag{2}$$

We can write $M = UU^\top$ for some $U \in \mathbb{R}^{d \times r}$ to reparameterize the objective function. Let

$$f_i(U) := \frac{1}{2} \left( \langle UU^\top, A_i \rangle - y_i \right)^2. \tag{3}$$

We can compute

$$\bar{f}(U) := \underset{(A_i, y_i) \sim \mathcal{G}_0}{\mathbb{E}} f_i(U) = \frac{1}{2} \operatorname{Var}\langle UU^\top - M^*, A_i \rangle = \frac{1}{2} \left\| UU^\top - M^* \right\|_F^2. \tag{4}$$

We seek to solve the following optimization problem under the corruption model in Definition 1.2:

$$\min_{U \in \mathbb{R}^{d \times r}} \bar{f}(U). \tag{5}$$

The gradient Lipschitz constant and Hessian Lipschitz constant of $\bar{f}$ are given by the following result.

**Fact 3.4** ([Jin+17], Lemma 6). *For any $\Gamma > \sigma_1^\star$, $\bar{f}(U)$ has gradient Lipschitz constant $L_g = 16\Gamma$ and Hessian Lipschitz constant $L_H = 24\Gamma^{\frac{1}{2}}$ inside the region $\{U : \|U\|_{\text{op}}^2 < \Gamma\}$.*

### 3.1.2 Global Convergence to an Approximate SOSP

In this section, we apply our framework Theorem 3.1 to obtain global convergence from an arbitrary initialization to an approximate SOSP, by providing problem-specific analysis to guarantee that both Assumption 1.4 and algorithmic assumptions (I) and (II) required by Theorem 3.1 are satisfied.

**Theorem 3.5** (Global Convergence to a SOSP). *Consider the noiseless setting as in Theorem 1.7 with $\sigma = 0$ and $\epsilon = O(1/(\kappa^3 r^3))$. Assume we have an arbitrary initialization $U_0$ inside $\{U : \|U\|_{\text{op}}^2 \leq \Gamma\}$. There exists a sample size $n = \widetilde{O}\left((d^2 r^2 + dr \log(\Gamma/\xi))/\epsilon\right)$ such that with probability at least $1 - \xi$, Algorithm A.1 initialized at $U_0$ outputs a $(\frac{1}{24}\sigma_r^{\star 3/2}, \frac{1}{3}\sigma_r^\star)$-approximate SOSP using at most $O\left(r^2 \kappa^3 \log(1/\xi)\right)$ calls to robust mean estimation subroutine (Algorithm A.2).*

*Proof of Theorem 3.5.* To apply Theorem 3.1, we verify Assumption 1.4 first. To verify (i), for all $U$ and $A_i$, $f_i(U) = \frac{1}{2}\left(\langle UU^\top, A_i \rangle - y_i\right)^2 \geq 0$, so $f^* = 0$ is a uniform lower bound. We verify (ii) in Appendix C.2: conceptually, by Fact 3.4, $\bar{f}$ is gradient and Hessian Lipschitz; both gradient and Hessian of $f_i$ are sub-exponential and concentrate around those of $\bar{f}$. To check (iii), we calculate the gradients and Hessians of $f_i$ in Appendix C.1.1 and bound their covariances from above in Appendix C.1.2 and C.1.3. The result is summarized in the following lemma. Note that the domain of the target function in Algorithm A.1 and Theorem 3.1 is the Euclidean space $\mathbb{R}^D$, so we vectorize $U$ and let $D = dr$. The gradient becomes a vector in $\mathbb{R}^{dr}$ and the Hessian becomes a matrix in $\mathbb{R}^{dr \times dr}$.

**Lemma 3.6** (Gradient and Hessian Covariance Bounds). *For all $U \in \mathbb{R}^{d \times r}$ with $\|U\|_{\text{op}}^2 \leq \Gamma$ and $f_i$ defined in Equation (3), it holds*

$$\left\| \operatorname{Cov}(\operatorname{vec}(\nabla f_i(U))) \right\|_{\text{op}} \leq 8 \left\| UU^\top - M^* \right\|_F^2 \|U\|_{\text{op}}^2 \leq 32 r^2 \Gamma^3 \tag{6}$$

$$\left\| \operatorname{Cov}(\operatorname{vec}(H_i)) \right\|_{\text{op}} \leq 16r \left\| UU^\top - M^* \right\|_F^2 + 128 \|U\|_{\text{op}}^4 \leq 192 r^3 \Gamma^2 \tag{7}$$

We proceed to verify the algorithmic assumptions in Theorem 3.1. For the assumption (I), we prove the following Lemma in Appendix C.2 to show that all iterates stay inside the bounded region in which we compute the covariance bounds.

**Lemma 3.7.** *All iterates of Algorithm A.1 stay inside the region $\{U : \|U\|_{\text{op}}^2 \leq \Gamma\}$.*

To verify Theorem 3.1 (II), we let $\epsilon_g = \frac{1}{32}\sigma_r^{\star 3/2}, \epsilon_H = \frac{1}{4}\sigma_r^\star$ and $\sigma_g = 8r\Gamma^{1.5}, \sigma_H = 16r^{1.5}\Gamma$. So if we assume $\epsilon = O(1/(\kappa^3 r^3))$, then for the absolute constant $c$ in Theorem 3.1 it holds that

$$\sigma_g \sqrt{\epsilon} \leq c\epsilon_g \qquad \sigma_H \sqrt{\epsilon} \leq c\epsilon_H.$$

Hence, Theorem 3.1 applies and Algorithm A.1 outputs an $(\epsilon_g, \epsilon_H)$-approximate SOSP with high probability in polynomial time. To bound the runtime, since $\bar{f}(U_0) = 1/2 \left\| U_0 U_0^\top - M^* \right\|_F^2 = O(r^2 \Gamma^2)$ for an arbitrary initialization $U_0$ with $\|U_0\|_{\text{op}}^2 < \Gamma$, the initial distance can be bounded by $O(r^2 \Gamma^2)$. Setting $L_g = 16\Gamma, L_H = 24\Gamma^{1/2}, \bar{f}(U_0) = O(r^2 \Gamma^2), f^* = 0$ and thus $C_\epsilon = O(\sigma_r^{\star 3}/\Gamma)$, Proposition 2.2 implies that Algorithm A.1 outputs a $(\frac{1}{24}\sigma_r^{\star 3/2}, \frac{1}{3}\sigma_r^\star)$-approximate second order stationary point $U_{SOSP}$ in $O(r^2 \kappa^3 \log(1/\xi))$ steps with high probability. $\qquad\square$

### 3.1.3 Local Linear Convergence

In this section, we describe a local search algorithm that takes a $(\frac{1}{24}\sigma_r^{\star 3/2}, \frac{1}{3}\sigma_r^\star)$-approximate second-order stationary point as its initialization and achieves exact recovery even in the presence of outliers.

---

**Algorithm 3.1:** Local Inexact Gradient Descent

---

**Data:** The initialization $U_{SOSP}$ is a $(\frac{1}{24}\sigma_r^{\star 3/2}, \frac{1}{3}\sigma_r^\star)$-approximate SOSP, corruption fraction is $\epsilon$,
     corrupted samples are $\{(A_i, y_i)\}_{i=1}^n$, target distance to optima is $\iota$
**Result:** $U$ that is $\iota$-close in Frobenius norm to some global minimum
1   $\eta = 1/\Gamma, U_0 = U_{SOSP}$
2   **for** $t = 0, 1, \ldots$ **do**
3      $\widetilde{g}_t := \mathbf{RobustMeanEstimation}(\{\nabla f_i(U_t)\}_{i=1}^n, 4\epsilon)$
4      $U_{t+1} \leftarrow U_t - \eta \widetilde{g}_t$

---

**Theorem 3.8** (Local Linear Convergence). *Consider the same noiseless setting as in Theorem 1.7. Assume we already found a $(\frac{1}{24}\sigma_r^{\star 3/2}, \frac{1}{3}\sigma_r^\star)$-approximate SOSP $U_{SOSP}$ of $\bar{f}$. Then there exists a sample size $n = \widetilde{O}(dr \log(1/\xi)/\epsilon)$ such that with probability at least $1 - \xi$, Algorithm 3.1 initialized at $U_{SOSP}$ outputs a solution that is $\iota$-close to some global minimum in Frobenius norm after $O(\kappa \log(\sigma_r^\star/\iota))$ calls to robust mean estimation subroutine (Algorithm A.2). Moreover, all iterates $U_t$ are $\frac{1}{3}\sigma_r^{\star 1/2}$-close to some global minimum in Frobenius norm.*

*Proof Sketch.* First we use known properties of $\bar{f} = \mathbb{E}_{(A_i, y_i) \sim \mathcal{G}_0} f_i$ from the literature [Jin+17] to show approximate SOSPs of $\bar{f}$ — in particular our initialization $U_{SOSP}$ — are in a small neighborhood of the global minima of $\bar{f}$. In that neighborhood, it was also known that $\bar{f}$ satisfies some local regularity conditions that enable gradient descent's linear convergence.

However, the algorithm only has access to the *inexact* gradient from the robust mean estimation subroutine (line 3 in Algorithm 3.1) and, therefore, we need to establish linear convergence of *inexact gradient descent*. We achieve this with the following iterative argument: As the iterate gets closer to global optima, the covariance bound of sample gradient in Equation (6) gets closer to 0. Because the accuracy of the robust mean estimation scales with the covariance bound (see Proposition 2.3), a more accurate estimate of population gradient $\nabla \bar{f}$ can be obtained via the robust estimation subroutine (Algorithm A.2). This, in turn, allows for an improved inexact gradient descent step, driving the algorithm towards an iterate that is even closer to global optima.

See Appendix C.3 for the complete proof.         □

## 4 Statistical Query Lower Bound for Low Rank Matrix Sensing

Our general algorithm (Theorem 3.1) leads to an efficient algorithm for robust low rank matrix sensing with sample complexity $O(d^2 r^2/\epsilon)$ (Theorem 3.3). Interestingly, the sample complexity of the underlying robust estimation problem — ignoring computational considerations — is $\Theta(dr/\epsilon)$. The information-theoretic upper bound of $O(dr/\epsilon)$ can be achieved by an exponential (in the dimension) time algorithm (generalizing the Tukey median to our regression setting); see, e.g., Theorem 3.5 in [Gao20].

Given this discrepancy, it is natural to ask whether the sample complexity achieved by our algorithm can be improved via a different computationally efficient method. In this section, we provide evidence that this may not be possible. In more detail, we establish a near-optimal information-computation tradeoff for the problem, within the class of Statistical Query (SQ) algorithms. To formally state our lower bound, we require basic background on SQ algorithms.

**Basics on SQ Model.** SQ algorithms are a class of algorithms that, instead of access to samples from some distribution $\mathcal{P}$, are allowed to query expectations of bounded functions over $\mathcal{P}$.

**Definition 4.1** (SQ Algorithms and STAT Oracle [Kea98]). Let $\mathcal{P}$ be a distribution on $\mathbb{R}^{d^2+1}$. A Statistical Query (SQ) is a bounded function $q : \mathbb{R}^{d^2+1} \to [-1, 1]$. For $\tau > 0$, the $\mathsf{STAT}(\tau)$ oracle

responds to the query $q$ with a value $v$ such that $|v - \mathbb{E}_{X \sim \mathcal{P}}[q(X)]| \leq \tau$. An SQ algorithm is an algorithm whose objective is to learn some information about an unknown distribution $\mathcal{P}$ by making adaptive calls to the corresponding $\mathsf{STAT}(\tau)$ oracle.

In this section, we consider $\mathcal{P}$ as the unknown corrupted distribution where $(A_i, y_i)$ are drawn. The SQ algorithm tries to learn the ground truth matrix $M^*$ from this corrupted distribution; the goal of the lower bound result is to show that this is hard.

The SQ model has the capability to implement a diverse set of algorithmic techniques in machine learning such as spectral techniques, moment and tensor methods, local search (e.g., Expectation Maximization), and several others [Fel+17]. A lower bound on the SQ complexity of a problem provides evidence of hardness for the problem. [Bre+21] established that (under certain assumptions) an SQ lower bound also implies a qualitatively similar lower bound in the low-degree polynomial testing model. This connection can be used to show a similar lower bound for low-degree polynomials.

Our main result here is a near-optimal SQ lower bound for robust low rank matrix sensing that applies even for rank $r = 1$, i.e., when the ground truth matrix is $M^* = uu^\top$ for some $u \in \mathbb{R}^d$. The choice of rank $r = 1$ yields the strongest possible lower bound in our setting because it is the easiest parameter regime: Recall that the sample complexity of our algorithm is $\widetilde{O}(d^2 r^2)$ as in Theorems 3.2 and 3.3, and the main message of our SQ lower bound is to provide evidence that the $d^2$ factor is necessary for computationally efficient algorithms *even if $r = 1$*.

**Theorem 4.2** (SQ Lower Bound for Robust Rank-One Matrix Sensing). *Let $\epsilon \in (0, 1/2)$ be the fraction of corruptions and let $c \in (0, 1/2)$. Assume the dimension $d \in \mathbb{N}$ is sufficiently large. Consider the $\epsilon$-corrupted rank-one matrix sensing problem with ground-truth matrix $M^* = uu^\top$ and noise $\sigma^2 = O(1)$. Any SQ algorithm that outputs $\widehat{u}$ with $\|\widehat{u} - u\| = O(\epsilon^{1/4})$ either requires $2^{\Omega(d^c)}/d^{2-4c}$ queries or makes at least one query to $\mathsf{STAT}\left(e^{O(1/\sqrt{\epsilon})}/O(d^{1-2c})\right)$.*

In other words, we show that, when provided with SQ access to an $\epsilon$-corrupted distribution, approximating $u$ is impossible unless employing a statistical query of higher precision than what can be achieved with a strictly sub-quadratic number (e.g., $d^{1.99}$) of samples. Note that the SQ oracle $\mathsf{STAT}(e^{O(1/\sqrt{\epsilon})}/O(d^{1-2c}))$ can be simulated with $O(d^{2-4c})/e^{O(1/\epsilon)}$ samples, and this bound is tight in general. Informally speaking, this theorem implies that improving the sample complexity from $d^2$ to $d^{2-4c}$ requires exponentially many queries. This result can be viewed as a near-optimal information-computation tradeoff for the problem, within the class of SQ algorithms.

The proof follows a similar analysis as in [DKS19; Dia+21], using one-dimensional moment matching to construct a family of corrupted conditional distributions, which induce a family of corrupted joint distributions that are SQ-hard to learn. We provide the details of the proof in Appendix D. Apart from the formal proof, in Appendix E we also informally discuss the intuition for why some simple algorithms that require $O(d)$ samples do not provide dimension-independent error guarantees.

## Acknowledgements

Shuyao Li was supported in part by NSF Awards DMS-2023239, NSF Award CCF-2007757 and the U. S. Office of Naval Research under award number N00014-22-1-2348. Yu Cheng was supported in part by NSF Award CCF-2307106. Ilias Diakonikolas was supported in part by NSF Medium Award CCF-2107079, NSF Award CCF-1652862 (CAREER), a Sloan Research Fellowship, and a DARPA Learning with Less Labels (LwLL) grant. Jelena Diakonikolas was supported in part by NSF Award CCF-2007757 and by the U. S. Office of Naval Research under award number N00014-22-1-2348. Rong Ge was supported in part by NSF Award DMS-2031849, CCF-1845171 (CAREER) and a Sloan Research Fellowship. Stephen Wright was supported in part by NSF Awards DMS-2023239 and CCF-2224213 and AFOSR via subcontract UTA20-001224 from UT-Austin.

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
