## Supplementary Material

Supplementary material is organized as follows. In Appendix A, we provide useful auxiliary facts and relevant technical results from previous works. Appendix B proves our result for general robust nonconvex optimization (Theorem 3.1). Appendix C provides omitted computation and proofs for robust low rank matrix sensing (Section 3.1). Appendix D proves our SQ lower bound (Section 4) for the sample complexity of efficient algorithms for the outlier-robust low rank matrix sensing problem, and Appendix E discusses the intuition why some simple algorithms that violate our SQ lower bound fail.

## A    Technical Preliminaries

### A.1    Notation and Auxiliary Facts

In the appendix, we use $\widetilde{O}(\cdot)$ notation to suppress, for conciseness, logarithmic dependences on all defined quantities, even if they do not appear inside $\widetilde{O}(\cdot)$. More precise statements can be found in the corresponding main parts of the paper. For matrix space $\mathbb{R}^{d \times r}$ and a set $\mathcal{X}^\star \subset \mathbb{R}^{d \times r}$, we use $\mathcal{P}_{\mathcal{X}^\star}(\cdot)$ to denote the Frobenius projection onto $\mathcal{X}^\star$, i.e., $\mathcal{P}_{\mathcal{X}^\star}(U) = \arg \min_{Z \in \mathcal{X}^\star} \|U - Z\|_F^2$. We use $\mathrm{dist}(U, \mathcal{X}^\star)$ to denote $\|U - \mathcal{P}_{\mathcal{X}^\star}(U)\|_F$.

**Fact A.1.** *For matrices $A, B$ with compatible dimensions,*

$$\|AB\|_F \leq \|A\|_{\mathrm{op}} \|B\|_F$$
$$\|AB\|_F \leq \|A\|_F \|B\|_{\mathrm{op}}$$

For two matrices $A$ and $B$, let $A \otimes B$ denote the Kronecker product of $A$ and $B$.

**Fact A.2.** *For matrices $A, B, C$ with compatible dimensions,* $\mathrm{vec}(ABC) = \left( C^\top \otimes A \right) \mathrm{vec}(B)$.

**Fact A.3.** $\|A \otimes B\|_{\mathrm{op}} = \|A\|_{\mathrm{op}} \|B\|_{\mathrm{op}}$

We will frequently use the following fact about the mean and the variance of the quadratic form for zero-mean multivariate normal distributions.

**Fact A.4.** *Let $X \sim \mathcal{N}(0, \Sigma)$ be a $k$-dimensional random variable and let $G$ be a $k \times k$ real matrix; then*

$$\mathbb{E}[X^\top G X] = \mathrm{tr}(G\Sigma) = \langle G, \Sigma \rangle$$
$$\mathrm{Var}[X^\top G X] = \mathrm{tr}(G\Sigma(G + G^\top)\Sigma)$$

### A.2    Simplified Proof for the Optimization Algorithm with Inexact Gradients and Hessians

---

**Algorithm A.1:** Nonconvex minimization with inexact gradients and Hessians [LW23]

---

1 **for** $t = 1, 2, \dots$ **do**
2      Obtain $\widetilde{g}_t$ from the inexact gradient oracle
3      **if** $\|\widetilde{g}_t\| > \epsilon_g$ **then**
4          $x_{t+1} = x_t - \frac{1}{L}\widetilde{g}_t$
5      **else**
6          Obtain $\widetilde{H}_t$ from the inexact Hessian oracle
7          Compute smallest eigenvalue and its corresponding eigenvector $(\widetilde{\lambda}_t, \widetilde{p}_t)$
8          **if** $\widetilde{\lambda}_t < -\epsilon_H$ **then**
9              $\sigma_t = \pm 1$ with probability $\frac{1}{2}$
10              $x_{t+1} = x_t + \frac{2\epsilon_H}{L_H}\sigma_t \widetilde{p}_t$
11          **else**
12              return $x_t$

---

We stated the following guarantee (Proposition 2.2) in Section 2:

**Proposition A.5.** *Suppose a function $f$ is bounded below by $f^* > -\infty$, has $L_g$-Lipschitz gradient and $L_H$-Lipschitz Hessian, and its inexact gradient and Hessian computation $\widetilde{g}_t$ and $\widetilde{H}_t$ satisfy $\|\widetilde{g}_t - \nabla f(x_t)\| \leq \frac{1}{3}\epsilon_g$ and $\|\widetilde{H}_t - \nabla^2 f(x_t)\|_{\mathrm{op}} \leq \frac{2}{9}\epsilon_H$. Then there exists an algorithm (Algorithm A.1) with the following guarantees:*

1. *(Correctness) If Algorithm A.1 terminates and outputs $x_n$, then $x_n$ is a $(\frac{4}{3}\epsilon_g, \frac{4}{3}\epsilon_H)$-approximate second-order stationary point.*

2. *(Runtime) Algorithm A.1 terminates with probability 1. Let $C_\epsilon := \min\left(\frac{\epsilon_g^2}{6L_g}, \frac{2\epsilon_H^3}{9L_H^2}\right)$. With probability at least $1 - \delta$, Algorithm A.1 terminates after $k$ iterations for*

$$k = O\Big(\frac{f(x_0) - f^*}{C_\epsilon} + \frac{L_H^2 L_g^2 \epsilon_g^2}{\epsilon_H^6}\log\Big(\frac{1}{\delta}\Big)\Big).$$

This section proves the correctness and a slightly weaker runtime guarantee in terms of the dependence on $\delta$. We prove a $O(1/\delta)$ dependence instead of $O(\log(1/\delta))$, i.e., with probability at least $1 - \delta$, Algorithm A.1 terminates after $k$ iterations for

$$k = O\Big(\frac{f(x_0) - f^*}{\delta C_\epsilon}\Big). \tag{8}$$

*Proof of correctness and a weaker runtime [LW23].* We prove the correctness first. From the stopping criteria, we have $\|\widetilde{g}_k\| \leq \epsilon_g$ and $\widetilde{\lambda}_k \geq -\epsilon_H$. Thus,

$$\|\nabla f(x_k)\| \leq \|g_k\| + \frac{1}{3}\epsilon_g = \frac{4}{3}\epsilon_g.$$

To bound $\lambda_{\min}(\nabla^2 f(x_k))$, write $U_k = \widetilde{H}_k - \nabla^2 f(x_k)$. By Hessian inexactness condition, $\lambda_{\min}(-U_k) \geq -\|-U_k\|_{\mathrm{op}} \geq -\frac{2}{9}\epsilon_H$.

We use Weyl's theorem to conclude that

$$\lambda_{\min}(\nabla^2 f(x_k)) \geq \lambda_{\min}(\widetilde{H}_k) + \lambda_{\min}(-U_k) \geq -\epsilon_H - \frac{2}{9}\epsilon_H > -\frac{4}{3}\epsilon_H,$$

as required.

Now we analyze the runtime. We proceed by first establishing an expectation bound and then use Markov's inequality.

Write $v_t := \widetilde{g}_t - \nabla f(x_t)$. When the algorithm takes a gradient step, we have $\|\widetilde{g}_t\| > \epsilon_g$. By gradient inexactness condition, we have $\|v_t\| \leq \frac{1}{3}\|\widetilde{g}_t\|$, so that $\|\nabla f(x_t)\| \geq \frac{2}{3}\|\widetilde{g}_t\|$. From Taylor's Theorem, we have

$$
\begin{aligned}
f(x_{t+1}) &= f\left(x_t - \frac{1}{L_g}\widetilde{g}_t\right) \\
&\leq f(x_t) - \frac{1}{L_g}\nabla f(x_t)^\top \widetilde{g}_t + \frac{L_g}{2}\cdot\frac{1}{L_g^2}\|\widetilde{g}_t\|^2 \\
&= f(x_t) - \frac{1}{2L_g}\left(\|\nabla f(x_t)\|^2 - \|v_t\|^2\right) \\
&\leq f(x_t) - \frac{1}{2L_g}\left(\frac{4}{9}\|\widetilde{g}_t\|^2 - \frac{1}{9}\|\widetilde{g}_t\|^2\right) \\
&\leq f(x_t) - \frac{1}{6L_g}\|\widetilde{g}_t\|^2 \\
&\leq f(x_t) - \frac{1}{6L_g}\epsilon_g^2, \tag{9}
\end{aligned}
$$

For the negative curvature step, recall that $\|\widetilde{H}_t - \nabla^2 f(x_t)\|_{\mathrm{op}} \leq \frac{2}{9}\epsilon_H$. It follows from Taylor's theorem that

$$
\begin{aligned}
f(x_{t+1}) &= f\left(x_t + \frac{2\epsilon_H}{L_H}\sigma_t\widetilde{p}_t\right) \\
&\leq f(x_t) + 2\frac{\epsilon_H}{L_H}\nabla f(x_t)^\top \sigma_t\widetilde{p}_t + \frac{1}{2}\frac{4\epsilon_H^2}{L_H^2}\widetilde{p}_t^\top\nabla^2 f(x_t)\widetilde{p}_t + \frac{L_H}{6}\frac{8\epsilon_H^3}{L_H^3} \\
&= f(x_t) + \frac{2\epsilon_H^2}{L_H^2}\left(\widetilde{p}_t^\top\nabla^2 f(x_t)\widetilde{p}_t + \frac{2}{3}\epsilon_H\right) + 2\frac{\epsilon_H}{L_H}\nabla f(x_t)^\top\sigma_t\widetilde{p}_t
\end{aligned}
$$

When the algorithm takes a negative curvature step, we have $\widetilde{\lambda}_t < -\epsilon_H < 0$, so by Hessian inexactness condition, we have $\|\widetilde{H}_t - \nabla^2 f(x_t)\| \leq \frac{2}{9}\epsilon_H \leq \frac{2}{9}|\widetilde{\lambda}_t|$. It follows from the definition of operator norm and Cauchy-Schwarz that $|\widetilde{p}_t^\top \widetilde{H}_t\widetilde{p}_t - \widetilde{p}_t^\top\nabla^2 f(x_t)\widetilde{p}_t| \leq \|\widetilde{H}_t\widetilde{p}_t - \nabla^2 f(x_t)\widetilde{p}_t\| \leq \frac{2|\widetilde{\lambda}_t|}{9}$, so

$$
\widetilde{p}_t^\top\nabla^2 f(x_t)\widetilde{p}_t \leq \frac{2}{9}|\widetilde{\lambda}_t| + p_t^\top \widetilde{H}_t\widetilde{p}_t = \frac{2}{9}|\widetilde{\lambda}_t| + (-|\widetilde{\lambda}_t|) = -\frac{7}{9}|\widetilde{\lambda}_t|.
$$

We thus have

$$
\begin{aligned}
f(x_{t+1}) &\leq f(x_t) + \frac{2\epsilon_H^2}{L_H^2}\left(\widetilde{p}_t^\top\nabla^2 f(x_t)\widetilde{p}_t + \frac{2}{3}\epsilon_H\right) + 2\frac{\epsilon_H}{L_H}\nabla f(x_t)^\top\sigma_t\widetilde{p}_t \\
&\leq f(x_t) + \frac{2\epsilon_H^2}{L_H^2}\left(-\frac{7}{9}|\widetilde{\lambda}_t| + \frac{2}{3}\alpha_t\right) + 2\frac{\epsilon_H}{L_H}\nabla f(x_t)^\top\sigma_t\widetilde{p}_t \\
&\leq f(x_t) - \frac{2\epsilon_H^3}{9L_H^2} + 2\frac{\epsilon_H}{L_H}\nabla f(x_t)^\top\sigma_t\widetilde{p}_t.
\end{aligned} \tag{10}
$$

We have the following result for expected stopping time of Algorithm A.1. Here the expectation is taken with respect to the random variables $\sigma_t$ used at the negative curvature iterations. For purposes of this and later results, we define

$$
C_\epsilon := \min\left(\frac{\epsilon_g^2}{6L_g}, \frac{2\epsilon_H^3}{9L_H^2}\right). \tag{11}
$$

Assuming Lemma A.6, the runtime guarantee in Proposition A.5 follows from Markov inequality. $\qquad\square$

**Lemma A.6.** *Consider the same setting as in Proposition A.5. Let $T$ denote the iteration at which Algorithm A.1 terminates. Then $T < \infty$ almost surely and*

$$
\mathbb{E}\, T \leq \frac{f(x_0) - f^*}{C_\epsilon}, \tag{12}
$$

*where $C_\epsilon$ is defined in (11).*

The proof of this result is given below. It constructs a supermartingale[1] based on the function value and uses a supermartingale convergence theorem and optional stopping theorem to obtain the final result. A similar proof technique is used in [Ber+22] but for a line-search algorithm. We collect several relevant facts about supermartingales before proving the result.

First, we need to ensure the relevant supermartingale is well defined even after the algorithm terminates, so that it is possible to let the index $t$ of the supermartingale go to $\infty$.

**Fact A.7** ([Dur19, Theorem 4.2.9]). *If $T$ is a stopping time and $X_t$ is a supermartingale, then $X_{\min(T,t)}$ is a supermartingale.*

The following supermatingale convergence theorem will be used to ensure the function value converges, so that the algorithm terminates with probability 1.

---

[1]A supermartingale with respect to filtration $\{\mathcal{G}_1, \mathcal{G}_2, \dots\}$ is a sequence of random variables $\{Y_1, Y_2, \dots\}$ such that for all $k \in \mathbb{Z}_+$, (i) $\mathbb{E}\,|Y_t| < \infty$, (ii) $Y_t$ is $\mathcal{G}_t$-measurable, and (iii) $\mathbb{E}(Y_{t+1}\,|\mathcal{G}_t) \leq Y_t$.

**Fact A.8** (Supermartingale Convergence Theorem, [Dur19, Theorem 4.2.12]). *If $X_t \geq 0$ is a supermartingale, then as $t \to \infty$, there exists a random variable $X$ such that $X_t \to X$ a.s. and $\mathbb{E} X \leq \mathbb{E} X_0$.*

Finally, we will use the optional stopping theorem to derive the expected iteration complexity. Note that we use a version of the optional stopping theorem specific to nonnegative supermartingales that does not require uniform integrability.

**Fact A.9** (Optional Stopping Theorem, [Dur19, Theorem 4.8.4]). *If $X_n$ is a nonnegative supermartingale and $N \leq \infty$ is a stopping time, then $\mathbb{E} X_0 \geq \mathbb{E} X_N$.*

*Proof of Lemma A.6.* We first construct a supermartingale based on function values. Since $\mathbb{E}[\sigma_t] = 0$, linearity of expectation implies that $\mathbb{E}\left[2\frac{\alpha_t}{M}\nabla f(x_t)^\top \sigma_t \widetilde{p}_t | x_t\right] = 0$. We therefore have from (10) that

$$\mathbb{E}\left[f(x_{t+1}) | x_t\right] \leq f(x_t) - \frac{2\epsilon_H^3}{9L_H^2}.$$

By combining with the (deterministic) first-order decrease estimate (9), we have

$$\mathbb{E}\left[f(x_{t+1}) | x_t\right] \leq f(x_t) - \min\left(\frac{\epsilon_g^2}{6L_g}, \frac{2\epsilon_H^3}{9L_H^2}\right) = f(x_t) - C_\epsilon.$$

Consider the stochastic process $M_t := f(x_t) + tC_\epsilon$. We have

$$\begin{aligned}
\mathbb{E}\left[M_{t+1} | x_t\right] &= \mathbb{E}\left[f(x_{t+1}) + (t+1)C_\epsilon | x_t\right] \\
&\leq \mathbb{E}\left[f(x_t) - C_\epsilon + (t+1)C_\epsilon | x_t\right] \\
&= \mathbb{E}\left[f(x_t) + tC_\epsilon | x_t\right] = M_t.
\end{aligned}$$

We need to select a filtration to define the supermartingale $M_t$. We view $x_t$ as random variables defined with respect to $\sigma_i, i \leq k$. Since $M_t$ is expressed as a function of $x_t$ only, we define the filtration $\{\mathcal{G}_t\}$ to be the filtration generated by $x_t$, and it naturally holds that $M_t$ is $\mathcal{G}_t$-measurable for all $k$ and $\mathbb{E}\left[M_{t+1} | \mathcal{G}_t\right] = \mathbb{E}\left[M_{t+1} | x_t\right]$. Hence, $\{M_t\}$ is a supermartingale with respect to filtration $\{\mathcal{G}_t\}$.

Let $T$ denote the iteration at which our algorithm stops. Since the decision to stop at iteration $t$ depends only on $x_t$, we have $\{T = t\} \in \mathcal{G}_t$, which implies $T$ is a stopping time.

We will use the supermartingale convergence theorem (Fact A.8) to show that $T < +\infty$ almost surely, since the function value cannot decrease indefinitely as it is bounded by $f^*$ from below. To apply Fact A.8, we need to let $t \to \infty$, so we need to transform $\{M_t\}$ to obtain a supermartingale $\{Y_t\}$ that is well defined even after the algorithm terminates.

It follows from Fact A.7 that $Y_t := M_{\min(t,T)}$ is also a supermartingale. Since $Y_t \geq f^*$, it follows from the supermartingale convergence theorem (Fact A.8) applied to $Y_t - f^*$ that $Y_t \to Y_\infty$ almost surely for some random variable $Y_\infty$ with $\mathbb{E} Y_\infty \leq \mathbb{E} Y_0 = \mathbb{E} M_0 = f(x_0) < \infty$. Hence $\mathbb{P}[Y_\infty = +\infty] = 0$. On the other hand, as $t \to \infty$, we have $tC_\epsilon \to \infty$, so $T = +\infty \implies Y_t = M_t \geq f^* + tC_\epsilon \to \infty \implies Y_\infty = +\infty$. Therefore we have $\mathbb{P}[T < +\infty] = 1$.

We can then apply the optional stopping theorem (Fact A.9) to $Y_t - f^*$. It follows that

$$f^* + \mathbb{E} T \cdot C_\epsilon \leq \mathbb{E} f(x_T) + \mathbb{E} T \cdot C_\epsilon = \mathbb{E}[M_T] = \mathbb{E}[Y_T] \leq \mathbb{E}[Y_0] = \mathbb{E}[M_0] = f(x_0),$$

where the first equality uses $T < +\infty$ almost surely and the last inequality is Fact A.9. By reorganizing this bound, we obtain $\mathbb{E} T \leq \frac{f(x_0) - f^*}{C_\epsilon}$, as desired. $\qquad \square$

### A.3 Robust Mean Estimation—Omitted background

A class of algorithms that robustly estimate quantities in the presence of outliers under strong contamination model (Definition 1.1) are based on the stability of samples:

**Definition A.10** (Definition of Stability [Dia+17]). Fix $0 < \epsilon < 1/2$ and $\delta > \epsilon$. A finite set $S \subset \mathbb{R}^k$ is $(\epsilon, \delta)$-stable with respect to mean $\mu \in \mathbb{R}^k$ and $\sigma^2$ if for every $S' \subset S$ with $|S'| \geq (1 - \epsilon)|S|$, the following conditions hold: (i) $\|\mu_{S'} - \mu\| \leq \sigma\delta$, and (ii) $\left\|\bar{\Sigma}_{S'} - \sigma^2 I\right\|_{\text{op}} \leq \sigma^2\delta^2\epsilon$, where $\mu_{S'}$ and $\bar{\Sigma}_{S'}$ denote the empirical mean and empirical covariance over the set $S'$ respectively.

---
**Algorithm A.2:** RobustMeanEstimation with unknown covariance bound
---
**Data:** $0 < \epsilon < 1/2$ and $T$ is an $\epsilon$-corrupted set

**Result:** $\widehat{\mu}$ with $\|\widehat{\mu} - \mu_S\| = O(\sqrt{\|\Sigma\|_{\mathrm{op}}\,\epsilon})$

1   Initialize a weight function $w : T \to \mathbb{R}_{\geq 0}$ with $w(x) = 1/|T|$ for all $x \in T$;

2   **while** $\|w\|_1 \geq 1 - 2\epsilon$ **do**

3     $\mu(w) := \frac{1}{\|w\|_1} \sum_{x \in T} w(x)x$;

4     $\Sigma(w) := \frac{1}{\|w\|_1} \sum_{x \in T} w(x)(x - \mu(w))(x - \mu(w))^\top$;

5     Compute the largest eigenvector $v$ of $\Sigma(w)$ ;

6     $g(x) := |v^\top(x - \mu(w))|^2$;

7     Find the largest threshold $t$ such that $\Sigma_{x \in T: g(x) \geq t} w(x) \geq \epsilon$;

8     $f(x) := g(x)\mathbb{1}\{g(x) \geq t\}$ ;

9     $w(x) \leftarrow w(x)\left(1 - \frac{f(x)}{\max_{y \in T: w(y) \neq 0} f(y)}\right)$ ;

10   **return** $\mu(w)$
---

On input the corrupted version of a stable set, Algorithm A.2 has the following guarantee.

**Proposition A.11** (Robust Mean Estimation with Stability [DKP20, Theorem A.3]). *Let $T \subset \mathbb{R}^k$ be an $\epsilon$-corrupted version of a set $S$, where $S$ is $(C\epsilon, \delta)$-stable with respect to $\mu_S$ and $\sigma^2$, and where $C > 0$ is a sufficiently large constant. Algorithm A.2 on input $\epsilon$ and $T$ (but not $\sigma$ or $\delta$) returns (deterministically) a vector $\widehat{\mu}$ in polynomial time so that $\|\mu_S - \widehat{\mu}\| = O(\sigma\delta)$.*

Proposition A.11 requires the uncorrupted samples to be stable. With a large enough sample size, most independent and identically distributed samples from a bounded covariance distribution are stable. The remaining samples can be treated as corruptions.

**Proposition A.12** (Sample Complexity for Stability [DKP20, Theorem 1.4]). *Fix any $0 < \xi < 1$. Let $S$ be a multiset of $n$ independent and identically distributed samples from a distribution on $\mathbb{R}^k$ with mean $\mu$ and covariance $\Sigma$. Then, with probability at least $1 - \xi$, there exists a sample size $n = O\left(\frac{k \log k + \log(1/\xi)}{\epsilon}\right)$ and a subset $S' \subseteq S$ such that $|S'| \geq (1 - \epsilon)n$ and $S'$ is $(2\epsilon, \delta)$-stable with respect to $\mu$ and $\|\Sigma\|_{\mathrm{op}}$, where $\delta = O(\sqrt{\epsilon})$.*

*Proof of Proposition 2.3.* Proposition A.12 implies that for i.i.d. samples from a $k$-dimensional bounded covariance distribution to contain a stable set of more than $(1 - \epsilon)$-fraction of samples with high probability, we need $\widetilde{O}\left(\frac{k}{\epsilon}\right)$ samples. We refer to the remaining $\epsilon$-fraction of samples as unstable samples. Since the adversary corrupts an $\epsilon$-fraction of clean samples $S$, the input set $T$ can be considered as a $2\epsilon$-corrupted version of a stable set, if we view the unstable samples as corruptions. Therefore, Proposition A.11 applies and gives the desired error guarantee. $\qquad\square$

## B   Omitted Proofs in General Robust Nonconvex Optimization

**Sample Size for Successful Robust Mean Estimation**   For each iteration $t$ in Algorithm A.1 and Algorithm 3.1, we would like to robustly estimate the mean of a set of corrupted points $\{\mathrm{vec}\,(\nabla f(x_t))\}_{i=1}^n$ and/or $\{\mathrm{vec}\,(\nabla^2 f(x_t))\}_{i=1}^n$ where the inliers are drawn from a distribution with bounded covariance $\Sigma \preceq \sigma^2 I$. For fixed $x_t$, inliers are drawn independently and identically distributed (i.i.d.), but the dependence across all iterations $t$ is allowed.

**Theorem B.1.** *Let $\mathcal{X} \subset \mathbb{R}^l$ be a closed and bounded set with radius at most $\gamma$ (i.e., $\|x\| \leq \gamma$, $\forall x \in \mathcal{X}$). Let $p^*$ be a distribution over functions $h : \mathcal{X} \to \mathbb{R}^k$. Let $\xi \in (0, 1)$ be the failure probability. Suppose $h$ is $L$-Lipschitz and uniformly bounded, i.e., there exists $B > 0$ such that $\|h(x)\| \leq B$ $\forall x \in \mathcal{X}$ almost surely. Assume further that for each $x \in \mathcal{X}$ we have $\|\mathrm{Cov}_{h \sim p^*}(h(x))\|_{\mathrm{op}} \leq \sigma^2$. Let $S$ be a multiset of $n$ i.i.d. samples $\{h_i\}_{i=1}^n$ from $p^*$. Then there exists*

*a sample size*

$$n = O\left(\frac{k \log k + l \log\left(\frac{\gamma LB}{\sigma \epsilon \xi}\right)}{\epsilon}\right)$$

*such that with probability $1 - \xi$, it holds that for each $x \in \mathcal{X}$, there exists a subset $S' \subset S$ with $|S'| \geq (1 - 2\epsilon)n$ (potentially different subsets $S'$ for different $x$) such that $\{h_i(x)\}_{i \in S'}$ is $(2\epsilon, \delta)$-stable with respect to $\mu$ and $\sigma^2$, where $\delta = O(\sqrt{\epsilon})$.*

To prove this theorem, we work with an easier version of stability condition specialized to samples from a bounded covariance distribution.

**Claim B.2** (Claim 2.1 in [DKP20]). *(Stability for bounded covariance) Let $R \subset \mathbb{R}^k$ be a finite multiset such that $\|\mu_R - \mu\| \leq \sigma \delta$, and $\|\bar{\Sigma}_R - \sigma^2 I\|_{\text{op}} \leq \sigma^2 \delta^2 / \epsilon$ for some $0 \leq \epsilon \leq \delta$. Then $R$ is $(\Theta(\epsilon), \delta')$ stable with respect to $\mu$ and $\sigma^2$, where $\delta' = O(\delta + \sqrt{\epsilon})$.*

*Proof of Theorem B.1.* Given Claim B.2, it suffices to show that with probability $1 - \xi$, for each $x \in \mathcal{X}$ there exists a subset $S' \subset S$ with $|S'| \geq (1 - 2\epsilon)n$ such that

$$\left\| \mathop{\mathbb{E}}_{i \in S'} h_i(x) - \mathop{\mathbb{E}}_{h \sim p^*} h(x) \right\| \leq O(\sigma \delta) \tag{13}$$

$$\left\| \mathop{\mathbb{E}}_{i \in S'} (h_i(x) - \mathop{\mathbb{E}}_{h \sim p^*} h(x))(h_i(x) - \mathop{\mathbb{E}}_{h \sim p^*} h(x))^\top - \sigma^2 I \right\|_{\text{op}} \leq O(\sigma^2 \delta^2 / \epsilon) \tag{14}$$

By Proposition A.12, for each $x \in \mathcal{X}$, with probability $1 - \left(\frac{\xi \gamma BL}{\sigma \sqrt{\epsilon}}\right)^l \geq 1 - \xi / \left(\frac{\gamma BL}{\sigma \sqrt{\epsilon}}\right)^l$, there exists a subset $S' \subset S$ with $|S'| \geq (1 - 2\epsilon)n$ such that (13) and (14) hold.

We proceed with a net argument. Up to a multiplicative error that can be suppressed by $O(\cdot)$, if Equation (13) holds for some $x \in \mathcal{X}$, it also holds for all other $x'$ in a ball of radius $\sigma \delta / L$ because $h(\cdot)$ is $L$-Lipschitz. Similarly, (14) is equivalent to

$$\left| \mathop{\mathbb{E}}_{i \in S'} ((h_i(x) - \mathop{\mathbb{E}}_{h \sim p^*} h(x))^\top v)^2 - \sigma^2 v^\top v \right| \leq O(\sigma^2 \delta^2 / \epsilon) \qquad \forall v \in \mathbb{R}^k.$$

Since $h(\cdot)$ is $L$-Lipschitz and uniformly bounded by $B$, if (14) holds for some $x \in \mathcal{X}$, it also holds for all $x'$ in a ball of radius $\frac{\sigma^2 \delta^2}{\epsilon BL}$. Therefore, it suffices for Equation (13) and (14) to hold for a $\tau$-net of $\mathcal{X}$, where for $\delta = O(\sqrt{\epsilon})$ we have $\tau = \min\left(\frac{\sigma \delta}{L}, \frac{\sigma^2 \delta^2}{\epsilon BL}\right) = \Omega\left(\frac{\sigma \sqrt{\epsilon}}{BL}\right)$. An $\Omega\left(\frac{\sigma \sqrt{\epsilon}}{BL}\right)$-net of $\gamma$-radius ball in $\mathbb{R}^l$ has size $O\left(\left(\frac{\gamma BL}{\sigma \sqrt{\epsilon}}\right)^l\right)$. Taking a union bound over this net completes the proof. $\square$

**Proof of Main Theorem** We establish a slightly enhanced version of Theorem 3.1 by including the last sentence as an additional component to the original theorem.

**Theorem B.3.** *Suppose we are given $\epsilon$-corrupted set of functions $\{f_i\}_{i=1}^n$ for sample size $n$, generated according to Definition 1.2. Suppose Assumption 1.4 holds in a bounded region $\mathcal{B} \subset \mathbb{R}^D$ of radius $\Gamma$ with gradient and Hessian covariance bound $\sigma_g$ and $\sigma_H$ respectively, and we have an arbitrary initialization $x_0 \in \mathcal{B}$. For some $n = \widetilde{O}(D^2 / \epsilon)$, Algorithm A.1 initialized at $x_0$ outputs an $(\epsilon_g, \epsilon_H)$-approximate SOSP in polynomial time with high probability if the following conditions hold:*

*(I) All iterates $x_t$ in Algorithm A.1 stay inside the bounded region $\mathcal{B}$.*

*(II) For an absolute constant $c > 0$, it holds that $\sigma_g \sqrt{\epsilon} \leq c\epsilon_g$ and $\sigma_H \sqrt{\epsilon} \leq c\epsilon_H$.*

*Moreover, there exists an absolute constant $C_{est}$ such that for each iteration $t$, the gradient oracle $\widetilde{g}_t$ and Hessian oracle $\widetilde{H}_t$ satisfy $\|\widetilde{g}_t - \nabla f(x_t)\| \leq C_{est} \sigma_g \sqrt{\epsilon}$ and $\|\widetilde{H}_t - \nabla^2 f(x_t)\|_F \leq C_{est} \sigma_H \sqrt{\epsilon}$.*

And we recall that the we construct the gradient and Hessian oracles in Algorithm A.1 in the following way:

$$\widetilde{g}_t \leftarrow \textbf{RobustMeanEstimation}(\{\nabla f_i(x_t)\}_{i=1}^n, 4\epsilon)$$
$$\widetilde{H}_t \leftarrow \textbf{RobustMeanEstimation}(\{\nabla^2 f_i(x_t)\}_{i=1}^n, 4\epsilon)$$

If we ignore that the dependence between $\{\nabla f_i(x_t)\}$ introduced via $x_t$, this theorem follows directly from Proposition 2.3. Here we fix this technicality via a union bound over all $x_t$ with Theorem B.1.

*Proof.* By Proposition 2.2, it suffices to check that gradient and Hessian inexactness condition is satisfied. That is, for all iterations $t$, it holds that

$$\|\widetilde{g}_t - \nabla f(x_t)\| \leq \frac{1}{3}\epsilon_g, \tag{15}$$

$$\|\text{vec}(\widetilde{H}_t) - \text{vec}(\nabla^2 f(x_t))\| \leq \frac{2}{9}\epsilon_H. \tag{16}$$

Here we use the fact that $\|\text{vec}(\widetilde{H}_t) - \text{vec}(\nabla^2 f(x_t))\| = \|\widetilde{H}_t - \nabla^2 f(x_t)\|_F \geq \|\widetilde{H}_t - \nabla^2 f(x_t)\|_{\text{op}}$.

We proceed to apply Theorem B.1 and consider the function $h_g(x, A) = \text{vec}(\nabla_x f(x, A))$ and $h_H(x) = \text{vec}(\nabla^2_{xx} f(x, A))$. We consider $h_H : \mathbb{R}^D \to \mathbb{R}^{D^2}$ first. By assumption (I), all iterates never leave the bounded region $\mathcal{B}$ with high probability; we condition on this event in the remaining analysis and it follows that $\left\|\text{Cov}_{A \sim \mathcal{G}}(\text{vec}(\nabla^2 f(x_t, A)))\right\|_{\text{op}} \leq \sigma_H^2$ for all iterations $t$.

Assumption 1.4 (ii) posits that with high probability $h_H(\cdot, A)$ is $L_{D_H}$-Lipschitz and its $\ell_2$-norm is bounded above by $B_{D_H}$. We further condition on this event. Let $\xi \in (0, 1)$. Since $\|\text{Cov}_{A \in \mathcal{G}}(h_H(x, A))\|_{\text{op}} \leq \sigma_H$ by Assumption 1.4 (iii), there exists a sample size

$$n = O\left(\frac{D^2 \log D + D \log\left(\frac{\gamma L_{D_H} B_{D_H}}{\sigma_H \epsilon \xi}\right)}{\epsilon}\right),$$

such that with probability $1 - \xi$, it holds that for each $x \in \mathcal{X}$, there exists a $(2\epsilon, O(\sqrt{\epsilon}))$-stable subset of size at least $(1 - 2\epsilon)n$ in the sense of Definition A.10 (potentially different subsets $S'$ for different $x$). Therefore, conditioning on the event that for all $x$ there exists a stable subset, Proposition A.11 implies that there exists an absolute constant $C_{est}$ such that $\|\text{vec}(\widetilde{H}_t) - \text{vec}(\nabla^2 f(x_t))\| \leq C_{est}\sigma_H\sqrt{\epsilon}$ for all $t$. By assumption (II) with a sufficiently small constant $c$, we have $C_{est}\sigma_H\sqrt{\epsilon} \leq 2\epsilon_H/9$, and therefore $\|\text{vec}(\widetilde{H}_t) - \text{vec}(\nabla^2 f(x_t))\| \leq \frac{2}{9}\epsilon_H$.

Now we established that for Equation (16) to hold for all iterations $t$ with high probability, the sample complexity is required to be at least $n = \widetilde{O}(D^2/\epsilon)$. A similar argument implies that $\widetilde{O}(D/\epsilon)$ samples are needed for Equation (15) to hold for all iterations $t$ with high probability, which is dominated by $\widetilde{O}(D^2/\epsilon)$. $\qquad\square$

## C  Omitted Proofs in Low Rank Matrix Sensing

This section provides omitted computation and proofs for robust low rank matrix sensing (Section 3.1), establishing Theorem 1.7. Appendix C.1 computes the gradient and Hessian of $f_i$ defined in Equation (3) and their covariance upper bounds under Definition 1.6 with noiseless measurements ($\sigma = 0$). Appendix C.2 proves the global convergence of our general robust nonconvex optimization framework (Theorem 3.1) applied to the noiseless robust low rank matrix sensing problem, which leads to an approximate SOSP, and Appendix C.3 proves the local linear convergence to a global minimum from this approximate SOSP. Finally, Appendix C.4 discusses the case of noisy measurements ($\sigma \neq 0$) under Definition 1.6.

### C.1  Omitted Computation in Low Rank Matrix Sensing

#### C.1.1  Sample Gradient and Sample Hessian

**Summary of Results**  Consider the noiseless setting as in Theorem 1.7 with $\sigma = 0$. We now compute the gradient and the Hessian of $f_i : \mathbb{R}^{d \times r} \to \mathbb{R}$. We firstly summarize the results. The gradient of $f_i$ at point $U$, denoted by $\nabla f_i(U)$, is a matrix in $\mathbb{R}^{d \times r}$ given by

$$\nabla f_i(U) = \langle UU^\top - M^*, A_i \rangle (A_i + A_i^\top)U.$$

The Hessian of $f_i$ at point $U$, denoted by $(H_i)_U$, is a linear operator $(H_i)_U : \mathbb{R}^{d \times r} \to \mathbb{R}^{d \times r}$. The operator $(H_i)_U$ acting on a matrix $Y \in \mathbb{R}^{d \times r}$ gives:

$$(H_i)_U(Y) = \langle UU^\top - M^*, A_i \rangle (A_i + A_i^\top) Y + \langle Y, (A_i + A_i^\top) U \rangle (A_i + A_i^\top) U.$$

We want to identify this linear map with a matrix $H_i(U)$ of dimension $\mathbb{R}^{dr \times dr}$. We vectorize both the domain and the codomain of $(H_i)_U$ so that it can be represented by a matrix.

Let $I_r$ and $I_d$ denote the identity matrices of dimension $r \times r$ and $d \times d$ respectively.

$$H_i(U) = \langle UU^\top - M^*, A_i \rangle I_r \otimes (A_i + A_i^\top) + \mathrm{vec}\left((A_i + A_i^\top) U\right) \mathrm{vec}\left((A_i + A_i^\top) U\right)^\top.$$

In this paper, we sometimes abuse the notation and use $\nabla^2 f_i(U)$ to refer to either $H_i(U)$ or $(H_i)_U$, but its precise meaning should be clear from its domain.

**Computation of Gradients and Hessians**    Let $(D^k f_i)_U$ denote the $k$-th order derivative of $f_i$ at point $U \in \mathbb{R}^{d \times r}$ and $\mathcal{L}(X, Y)$ denote the space of all linear mappings from $X$ to $Y$. We consider higher-order derivatives as linear maps:

$$(Df_i)_U : \mathbb{R}^{d \times r} \to \mathbb{R}$$
$$(D^2 f_i)_U : \mathbb{R}^{d \times r} \to \mathcal{L}(\mathbb{R}^{d \times r}, \mathbb{R})$$

We identify $(Df_i)_U$ with the matrix $\nabla f_i(U)$ such that

$$(Df_i)_U(Z) = \langle \nabla f_i(U), Z \rangle$$

and identify $(D^2 f_i)_U$ with a linear operator $(H_i)_U : \mathbb{R}^{d \times r} \to \mathbb{R}^{d \times r}$ such that

$$(D^2 f_i)_U(Y)(Z) = \langle (H_i)_U(Y), Z \rangle.$$

Since $f_i$ is differentiable, applying its derivative at $U$ to a matrix $Z$ gives the corresponding directional derivative at $U$ for the direction $Z$.

$$
\begin{aligned}
(Df_i)_U(Z) &= \frac{\mathrm{d}}{\mathrm{d}t} f_i(U + tZ)\Big|_{t=0} \\
&= \frac{\mathrm{d}}{\mathrm{d}t} \frac{1}{2} (\langle (U + tZ)(U + tZ)^\top, A_i \rangle - y_i)^2 \Big|_{t=0} \\
&= (\langle UU^\top, A_i \rangle - y_i) \left( \frac{\mathrm{d}}{\mathrm{d}t} \langle UU^\top + t^2 ZZ^\top + t(ZU^\top + UZ^\top), A_i \rangle \right)\Big|_{t=0} \\
&= (\langle UU^\top, A_i \rangle - y_i) \langle \frac{\mathrm{d}}{\mathrm{d}t} \left( t^2 ZZ^\top + t(ZU^\top + UZ^\top) \right), A_i \rangle \Big|_{t=0} \\
&\overset{(a)}{=} (\langle UU^\top, A_i \rangle - y_i) \langle ZU^\top + UZ^\top, A_i \rangle \\
&= (\langle UU^\top, A_i \rangle - y_i) \langle A_i, ZU^\top \rangle + \langle A_i, UZ^\top \rangle \\
&= (\langle UU^\top, A_i \rangle - y_i) \langle A_i U, Z \rangle + \langle U^\top A_i, Z^\top \rangle \\
&\overset{(b)}{=} (\langle UU^\top, A_i \rangle - y_i) \langle (A_i + A_i^\top) U, Z \rangle.
\end{aligned}
$$

From (a) we conclude $(Df_i)_U$ is the functional $Z \mapsto (\langle UU^\top, A_i \rangle - y_i) \langle ZU^\top + UZ^\top, A_i \rangle$; it would be easier to calculate the second derivative from this form. From (b), using $y_i = \langle A_i, M^* \rangle$, we obtain the following closed-form expression for the gradient $\nabla f_i(U)$:

$$\nabla f_i(U) = \langle UU^\top - M^*, A_i \rangle (A_i + A_i^\top) U.$$

To calculate the second derivative of $f_i$ at point $U$, we study the variation of its first derivative $U \mapsto (Df_i)_U$ at direction $Y$. Recall that the second derivative lives in the space of linear functionals

$\mathcal{L}(\mathbb{R}^{d\times r}, \mathbb{R})$ and we use $Z \in \mathbb{R}^{d\times r}$ to denote the input of this linear functional.

$$
\begin{aligned}
(D^2 f_i)_U(Y) &= \frac{\mathrm{d}}{\mathrm{d}t}(Df_i)_{U+tY}\Big|_{t=0} \\
&= \frac{\mathrm{d}}{\mathrm{d}t}\left\{ Z \mapsto \left( \langle (U+tY)(U+tY)^\top, A_i \rangle - y_i \right) \langle Z(U+tY)^\top + (U+tY)Z^\top, A_i \rangle \right\}\Big|_{t=0} \\
&= \left( \langle UU^\top, A_i \rangle - y_i \right) \frac{\mathrm{d}}{\mathrm{d}t}\left\{ Z \mapsto \langle Z(U+tY)^\top + (U+tY)Z^\top, A_i \rangle \right\}\Big|_{t=0} + \\
&\quad \frac{\mathrm{d}}{\mathrm{d}t}\left\{ \langle (U+tY)(U+tY)^\top, A_i \rangle - y_i \right\}\Big|_{t=0}\left\{ Z \mapsto \langle ZU^\top + UZ^\top, A_i \rangle \right\} \\
&= \left( \langle UU^\top, A_i \rangle - y_i \right)\left\{ Z \mapsto \langle ZY^\top + YZ^\top, A_i \rangle \right\} + \\
&\quad \langle UY^\top + YU^\top, A_i \rangle\left\{ Z \mapsto \langle ZU^\top + UZ^\top, A_i \rangle \right\} \\
&= Z \mapsto \left( \langle UU^\top, A_i \rangle - y_i \right)\langle ZY^\top + YZ^\top, A_i \rangle + \langle UY^\top + YU^\top, A_i \rangle\langle ZU^\top + UZ^\top, A_i \rangle \\
&= Z \mapsto (\langle UU^\top, A_i \rangle - y_i)\langle (A_i + A_i^\top)Y, Z \rangle + \langle Y, (A_i + A_i^\top)U \rangle\langle (A_i + A_i^\top)U, Z \rangle.
\end{aligned}
$$

Recall that we identify $(D^2 f_i)_U$ with a linear operator $(H_i)_U : \mathbb{R}^{d\times r} \to \mathbb{R}^{d\times r}$ such that

$$
(D^2 f_i)_U(Y)(Z) = \langle (H_i)_U(Y), Z \rangle.
$$

Note that $(H_i)_U : \mathbb{R}^{d\times r} \to \mathbb{R}^{d\times r}$ viewed as a fourth-order tensor is difficult to write down in a closed form, so we vectorize both the domain and the codomain of $(H_i)_U$ so that it can be represented by a matrix.

The operator $(H_i)_U$ acting on matrix $Y \in \mathbb{R}^{d\times r}$ gives

$$
(H_i)_U(Y) = \langle UU^\top - M^*, A_i \rangle(A_i + A_i^\top)Y + \langle Y, (A_i + A_i^\top)U \rangle(A_i + A_i^\top)U.
$$

We identify this linear map with a matrix $H_i(U)$ of dimension $\mathbb{R}^{dr\times dr}$, acting on the vectorized version of $Y$. As a shorthand, write $y = \mathrm{vec}(Y)$ and $B_i = A_i + A_i^\top$.

Let $I_r$ and $I_d$ denote the identity matrix of dimension $r \times r$ and $d \times d$ respectively. Then

$$
\begin{aligned}
H_i(U)\, y &= \mathrm{vec}\left( \langle UU^\top - M^*, A_i \rangle(A_i + A_i^\top)Y + \langle Y, (A_i + A_i^\top)U \rangle(A_i + A_i^\top)U \right) \\
&= \langle UU^\top - M^*, A_i \rangle\, \mathrm{vec}\left( B_i Y I_r \right) + \mathrm{vec}\left( B_i U \right)\mathrm{vec}(B_i U)^\top \mathrm{vec}(Y) \\
&= \langle UU^\top - M^*, A_i \rangle I_r \otimes B_i\, \mathrm{vec}(Y) + \mathrm{vec}(B_i U)\mathrm{vec}(B_i U)^\top \mathrm{vec}(Y) \\
&= \langle UU^\top - M^*, A_i \rangle(I_r \otimes B_i)y + \mathrm{vec}(B_i U)\mathrm{vec}(B_i U)^\top y.
\end{aligned}
$$

Hence

$$
H_i(U) = \langle UU^\top - M^*, A_i \rangle I_r \otimes B_i + \mathrm{vec}(B_i U)\,\mathrm{vec}(B_i U)^\top.
$$

### C.1.2 Gradient Covariance Bound

**Lemma C.1** (Equation (6)). *Consider the noiseless setting as in Theorem 1.7 with $\sigma = 0$. For all $U \in \mathbb{R}^{d\times r}$ with $\|U\|_{\mathrm{op}}^2 \leq \Gamma$, it holds that*

$$
\left\| \mathrm{Cov}(\mathrm{vec}(\nabla f_i(U))) \right\|_{\mathrm{op}} \leq 8\left\| UU^\top - M^* \right\|_F^2 \|U\|_{\mathrm{op}}^2 \leq 32 r^2 \Gamma^3. \tag{17}
$$

Recall that $\Gamma \geq 36\sigma_1^\star$, so we have the following trivial bound that is used frequently

$$
\|M^*\|_{\mathrm{op}} = \sigma_1^\star \leq \Gamma/36 < \Gamma.
$$

*Proof of Equation (6).* Recall that

$$
\nabla f_i(U) = \langle UU^\top - M^*, A_i \rangle(A_i + A_i^\top)U.
$$

We frequently encounter the following calculations.

**Lemma C.2.** *Let $P, Q \in \mathbb{R}^{d \times d}$ be given and let $A_i$ have i.i.d. standard Gaussian entries. Then*

$$\mathbb{E}[\langle P, A_i \rangle \langle Q, A_i \rangle] = \langle P, Q \rangle, \tag{18}$$

$$\mathrm{Var}[\langle P, A_i \rangle \langle Q, A_i \rangle] \le 2 \|P\|_F^2 \|Q\|_F^2. \tag{19}$$

*Proof.* By Definition 1.6, $X := \mathrm{vec}(A_i)$ is a standard Gaussian vector with identity covariance $\Sigma = I_{d^2}$. Let $a = \mathrm{vec}(P), b = \mathrm{vec}(Q)$, and $G = ab^\top$. Fact A.4 implies that

$$\mathbb{E}\langle P, A_i \rangle \langle Q, A_i \rangle = \langle ab^\top, I_{dr} \rangle = \langle a, b \rangle = \langle P, Q \rangle$$

$$\begin{aligned}
\mathrm{Var}[\langle P, A_i \rangle \langle Q, A_i \rangle] &= \mathrm{tr}(G\Sigma(G + G^\top)\Sigma) = \mathrm{tr}(G(G + G^\top)) \\
&= \mathrm{tr}(ab^\top(ab^\top + ba^\top)) \\
&= b^\top a\, \mathrm{tr}(ab^\top) + b^\top b\, \mathrm{tr}(aa^\top) \\
&= \langle a, b \rangle^2 + \|a\|^2 \|b\|^2 \\
&\le 2 \|a\|^2 \|b\|^2 \qquad \text{(Cauchy-Schwarz)} \\
&= 2 \|P\|_F^2 \|Q\|_F^2. \qquad \qquad \square
\end{aligned}$$

To compute the mean and the variance of the gradient $\nabla f_i(U)$, it is more convenient to work with $\langle \nabla f_i(U), Z \rangle$ for some $Z \in \mathbb{R}^{d \times r}$ with $\|Z\|_F = 1$.

$$\begin{aligned}
\langle \nabla f_i(U), Z \rangle &= \langle UU^\top - M^*, A_i \rangle \langle (A_i + A_i^\top)U, Z \rangle \\
&= \langle UU^\top - M^*, A_i \rangle \langle A_i + A_i^\top, ZU^\top \rangle \\
&= \langle UU^\top - M^*, A_i \rangle \langle ZU^\top + UZ^\top, A_i \rangle.
\end{aligned}$$

Lemma C.2 implies that the expectation

$$\begin{aligned}
\mathbb{E}\langle \nabla f_i(U), Z \rangle &= \langle UU^\top - M^*, ZU^\top + UZ^\top \rangle \\
&= \langle ((UU^\top - M^*) + (UU^\top - M^*)^\top) U, Z \rangle \\
&= \langle 2(UU^\top - M^*)U, Z \rangle.
\end{aligned}$$

By linearity of expectation, we conclude

$$\mathbb{E}\, \nabla f_i(U) = 2(UU^\top - M^*)U. \tag{20}$$

The variance bound can also be obtained via Lemma C.2

$$\mathrm{Var}\langle \nabla f_i(U), Z \rangle \le 2 \|UU^\top - M^*\|_F^2 \|ZU^\top + UZ^\top\|_F^2 \le 8\|UU^\top - M^*\|_F^2 \|U\|_{\mathrm{op}}^2 \|Z\|_F^2,$$

where the last inequality comes from Fact A.1.

Since $\mathrm{Var}\langle \nabla f_i(U), Z \rangle = \mathrm{Var}[\mathrm{vec}(\nabla f_i(U))^\top \mathrm{vec}(Z)] = \mathrm{vec}(Z)^\top \mathrm{Cov}(\mathrm{vec}(\nabla f_i(U))) \mathrm{vec}(Z)$, we have

$$\begin{aligned}
\|\mathrm{Cov}(\mathrm{vec}(\nabla f_i(U)))\|_{\mathrm{op}} &\le 8 \|UU^\top - M^*\|_F^2 \|U\|_{\mathrm{op}}^2 \\
&\le 8\Gamma (\|UU^\top\|_F + \|M^*\|_F)^2 \\
&\le 8\Gamma(r\Gamma + r\sigma_1^\star)^2 = 32r^2\Gamma^3. \qquad \square
\end{aligned}$$

### C.1.3 Hessian Covariance Bound

**Lemma C.3** (Equation (7)). *Consider the noiseless setting as in Theorem 1.7 with $\sigma = 0$. For all $U \in \mathbb{R}^{d \times r}$ with $\|U\|_{\mathrm{op}}^2 \le \Gamma$, it holds that*

$$\|\mathrm{Cov}(\mathrm{vec}(H_i))\|_{\mathrm{op}} \le 16r \|UU^\top - M^*\|_F^2 + 128 \|U\|_{\mathrm{op}}^4 \le 192r^3\Gamma^2. \tag{21}$$

*Proof.* As a shorthand, we write $B_i = A_i + A_i^\top$. Recall that
$$H_i(U) = \langle UU^\top - M^*, A_i \rangle I_r \otimes B_i + \text{vec}\,(B_i U)\,\text{vec}\,(B_i U)^\top.$$

Let $W$ be a $dr \times dr$ matrix that has Frobenius norm 1. We work with $\langle H_i(U), W \rangle$ as we did in the gradient covariance bound calculation.

By Young's inequality for $L^2$ random variables (i.e., $\|\mathcal{A} + \mathcal{B}\|_{L^2}^2 \le 2\|\mathcal{A}\|_{L^2}^2 + 2\|\mathcal{B}\|_{L^2}^2$),
$$\text{Var}\langle H_i(U), W \rangle \le 2 \underbrace{\text{Var}\left[\langle UU^\top - M^*, A_i\rangle\langle I_r \otimes B_i, W\rangle\right]}_{\mathcal{A}} + 2\underbrace{\text{Var}\left[\langle \text{vec}\,(B_i U)\,\text{vec}(B_i U)^\top, W\rangle\right]}_{\mathcal{B}}.$$

We first bound the term $\mathcal{A}$. Note $I_r \otimes B_i$ consists of $r$ copies of $B_i$ in the diagnal
$$I_r \otimes B_i = \begin{bmatrix} B_i & \cdots & O \\ \vdots & \ddots & \vdots \\ O & \cdots & B_i \end{bmatrix}.$$

We partition $W$ into $r^2$ submatrices of dimension $d \times d$.
$$W = \begin{bmatrix} W_{11} & \cdots & W_{1r} \\ \vdots & \ddots & \vdots \\ W_{r1} & \cdots & W_{rr} \end{bmatrix}.$$

Then
$$\text{Var}[\langle UU^\top - M^*, A_i\rangle\langle I_r \otimes B_i, W\rangle]$$
$$= \text{Var}[\langle UU^\top - M^*, A_i\rangle\langle B_i, \sum_{i=1}^r W_{ii}\rangle]$$
$$= \text{Var}[\langle UU^\top - M^*, A_i\rangle\langle A_i, \sum_{i=1}^r (W_{ii} + W_{ii}^\top)\rangle]$$
$$\le 2 \|UU^\top - M^*\|_F^2 \left\|\sum_{i=1}^r (W_{ii} + W_{ii}^\top)\right\|_F^2,$$

where the last inequality uses Lemma C.2.

Write $a_i = \text{vec}(W_{ii}) \in \mathbb{R}^{d^2}, b_i = \text{vec}(W_{ii}^\top), c_i = a_i + b_i$. Then $\|a_i\| = \|b_i\| = \|W_{ii}\|_F$ and
$$\left\|\sum_{i=1}^r (W_{ii} + W_{ii}^\top)\right\|_F^2 = \left\|\sum_{i=1}^r c_i\right\|^2$$
$$= \sum_{j=1}^{d^2} \left(\sum_{i=1}^r c_{ij}\right)^2 \le \sum_{j=1}^{d^2} r \sum_{i=1}^r c_{ij}^2 \qquad \text{(Cauchy-Schwarz)}$$
$$= r \sum_{i=1}^r \|c_i\|^2 = r \sum_{i=1}^r \|a_i + b_i\|^2$$
$$\le 2r \left(\sum_{i=1}^r (\|a_i\|^2 + \|b_i\|^2)\right)$$
$$= 4r \sum_{i=1}^r \|W_{ii}\|_F^2 \le 4r \|W\|_F^2 = 4r.$$

Therefore,
$$\text{Var}[\langle UU^\top - M^*, A_i\rangle\langle I_r \otimes B_i, W\rangle] \le 2 \|UU^\top - M^*\|_F^2 \left\|\sum_{i=1}^r (W_{ii} + W_{ii}^\top)\right\|_F^2$$
$$\le 8r \|UU^\top - M^*\|_F^2 \|W\|_F^2 = 8r \|UU^\top - M^*\|_F^2.$$

To bound the term $\mathcal{B}$, we apply Fact A.4 with $\Sigma = \mathrm{Cov}(\mathrm{vec}(B_i U)) \in \mathbb{R}^{dr \times dr}$:

$$\mathrm{Var}\left[\langle \mathrm{vec}\,(B_i U)\,\mathrm{vec}(B_i U)^\top, W\rangle\right]$$
$$= \mathrm{Var}\left[\mathrm{vec}(B_i U)^\top W \,\mathrm{vec}(B_i U)\right]$$
$$= \mathrm{tr}(W\Sigma(W + W^\top)\Sigma)$$
$$= \langle W\Sigma, \Sigma W\rangle + \langle W\Sigma, \Sigma W^\top\rangle$$
$$\leq \|W\Sigma\|_F \|\Sigma W\|_F + \|W\Sigma\|_F \|\Sigma W^\top\|_F$$
$$\leq \|W\|_F \|\Sigma\|_{\mathrm{op}} \|\Sigma\|_{\mathrm{op}} \|W\|_F + \|W\|_F \|\Sigma\|_{\mathrm{op}} \|\Sigma\|_{\mathrm{op}} \|W\|_F$$
$$= 2\|W\|_F^2 \|\Sigma\|_{\mathrm{op}}^2 = 2\|\Sigma\|_{\mathrm{op}}^2,$$

where the last inequality comes from Fact A.1.

To bound $\|\Sigma\|_{\mathrm{op}}^2$, we use Fact A.2 again. Since $\mathrm{vec}(B_i U) = \mathrm{vec}(I_d B_i U) = (U^\top \otimes I_d)\mathrm{vec}(B_i)$, we have

$$\mathrm{Cov}(\mathrm{vec}\,(B_i U)) = (U^\top \otimes I_d)\,\mathrm{Cov}(\mathrm{vec}(B_i))(U \otimes I_d). \tag{22}$$

Let $P \in \mathbb{R}^{d^2 \times d^2}$ be the permutation that maps $\mathrm{vec}(A_i)$ to $\mathrm{vec}(A_i^\top)$. Then

$$\mathrm{Cov}(\mathrm{vec}(B_i)) = \mathrm{Cov}(\mathrm{vec}(A_i) + \mathrm{vec}(A_i^\top))$$
$$\leq 2I_{d^2} + 2\,\mathrm{Cov}(\mathrm{vec}(A_i), \mathrm{vec}(A_i^\top)) \qquad \text{(Young's inequality)}$$
$$= 2I_{d^2} + 2\,\mathrm{Cov}(\mathrm{vec}(A_i), P\,\mathrm{vec}(A_i))$$
$$= 2I_{d^2} + 2P\,\mathrm{Cov}(\mathrm{vec}(A_i), \mathrm{vec}(A_i))$$
$$= 2I_{d^2} + 2P.$$

Since $P$ is a permutation, we have $\|\mathrm{Cov}(\mathrm{vec}(B_i))\|_{\mathrm{op}} \leq 4$.

It follows from Equation (22) and Fact A.3 that $\|\mathrm{Cov}(\mathrm{vec}(B_i U))\|_{\mathrm{op}} \leq 4\|U\|_{\mathrm{op}}^2$ and therefore

$$\mathrm{Var}\left[\langle \mathrm{vec}\,(B_i U)\,\mathrm{vec}(B_i U)^\top, W\rangle\right] \leq 64\|U\|_{\mathrm{op}}^4 \|W\|_F^2 = 64\|U\|_{\mathrm{op}}^4.$$

We conclude that for all $W \in \mathbb{R}^{dr \times dr}$,

$$\mathrm{Var}\langle H_i, W\rangle \leq 16r\|UU^\top - M^*\|_F^2 + 128\|U\|_{\mathrm{op}}^4,$$

hence

$$\|\mathrm{Cov}(\mathrm{vec}(H_i))\|_{\mathrm{op}} \leq 16r\|UU^\top - M^*\|_F^2 + 128\|U\|_{\mathrm{op}}^4 \leq 192r^3\Gamma^2,$$

where the last inequality uses $r \geq 1$ and $\|U\|_{\mathrm{op}}^2 \leq \Gamma$. $\qquad\square$

## C.2 Global Convergence in Low Rank Matrix Sensing—Omitted Proofs

We firstly verify Assumption 1.4 (ii). We recall the entire assumption below:

**Assumption C.4** (Assumption 1.4). *There exists a bounded region $\mathcal{B}$ such that the function $f$ satisfies:*

*(i) There exists a lower bound $f^* > -\infty$ such that for all $x \in \mathcal{B}$, $f(x, A) \geq f^*$ with probability $1$.*

*(ii) There exist parameters $L_{D_g}, L_{D_H}, B_{D_g}, B_{D_H}$ such that with high probability over the randomness in $A$, $f(\cdot, A)$ is $L_{D_g}$-gradient Lipschitz and $L_{D_H}$-Hessian Lipschitz, and the $\ell_2$-norm of gradient and the Frobenius norm of Hessian of $f(\cdot, A)$ are upper bounded by $B_{D_g}$ and $B_{D_H}$ respectively.*

*(iii) There exist parameters $\sigma_g, \sigma_H > 0$ such that*

$$\|\mathrm{Cov}_{A \sim \mathcal{G}}(\nabla f(x, A))\|_{\mathrm{op}} \leq \sigma_g \quad \text{and} \quad \|\mathrm{Cov}_{A \sim \mathcal{G}}(\mathrm{vec}(\nabla^2 f(x, A)))\|_{\mathrm{op}} \leq \sigma_H.$$

*Verifying Assumption 1.4(ii).* We take the bounded region $\mathcal{B}$ to be the region $\{U : \|U\|_{\mathrm{op}}^2 \leq \Gamma\}$. Let $f_i(U) = \frac{1}{2}\left(\langle UU^\top, A_i\rangle - y_i\right)^2$ be the cost function corresponding to clean samples. We need

to verify that $\nabla f_i(U)$ and $\nabla^2 f_i(U)$ are $\mathrm{poly}(dr\Gamma/\epsilon)$-Lipschitz and bounded within $\mathcal{B}$ with high probability so that the violated samples constitute at most an $\epsilon$-fraction.

We discuss gradient-Lipschitzness as an example, focusing on the constant $L_{D_g}$. The rest of conditions can be checked similarly. We drop the subscript $L := L_{D_g}$ in the following analysis for conciseness of the notation.

Since $\nabla f_i(X) - \nabla f_i(Y) = \langle XX^\top - M^*, A_i\rangle(A_i + A_i^\top)(X - Y) + \langle XX^\top - YY^\top, A_i\rangle(A_i + A_i^\top)Y$, it suffices that $\|A_i\|_F^2 \leq 2d^2 + 3/\epsilon$ for $\nabla f_i(\cdot)$ to be $\mathrm{poly}(dr\Gamma/\epsilon)$-Lipschitz. Since $\|A_i\|_F^2$ follows chi-square distribution with degree of freedom $d^2$, by Laurent-Massart bound, we have

$$\mathbb{P}\{\|A_i\|_F^2 - d^2 \geq d^2 + 3/\epsilon\} \leq \exp(-1/\epsilon).$$

This completes the verification of Assumption 1.4(ii) for $L_{D_g}$. We proceed to discuss why this high probability result implies that at most an $\epsilon$-fraction of clean samples violates gradient Lipschitzness. By Chernoff's inequality, the probability that more than $\epsilon$-fraction of uncorrupted $A_i$'s fail to satisfy $\|A_i\|_F^2 \leq 2d^2 + 3/\epsilon$ is less than

$$\exp\left(-n\left((1-\epsilon)\log\frac{1-\epsilon}{1-\exp(-1/\epsilon)} + \epsilon\log\frac{\epsilon}{\exp(-1/\epsilon)}\right)\right) = O\left(\frac{1}{\epsilon}\exp(-n)\right),$$

which is a small if $n = \widetilde{O}\left(\frac{1}{\epsilon}\right)$ is sufficiently large. $\square$

**Corollary C.5.** *Consider the noiseless setting as in Theorem 1.7 with $\sigma = 0$. There exists a sample size $n = \widetilde{O}\left(\frac{d^2 r^2}{\epsilon}\right)$ such that with $n$ samples, all subroutine calls to* RobustMeanEstimation *(Algorithm A.2) to estimate population gradients $\nabla \bar{f}(U_t)$ and population Hessians $\nabla^2 \bar{f}(U_t)$ succeed with high probability. In light of Equations (6) and (7), this means that there exists a large enough constant $C_{est}$ such that with high probability, for all iterations $t$, we have*

$$\left\|\widetilde{g}_t - \nabla \bar{f}(U_t)\right\|_F \leq 2C_{est}\left\|U_t U_t^\top - M^*\right\|_F \|U_t\|_{\mathrm{op}} \tag{23}$$

$$\left\|\widetilde{H}_t - \nabla^2 \bar{f}(U_t)\right\|_F \leq 4r^{1/2}\left\|U_t U_t^\top - M^*\right\|_F + 16\|U_t\|_{\mathrm{op}}^2. \tag{24}$$

*Proof.* This follows directly from the last sentence of Theorem B.3, with $D = d^2 r^2$ and $\sigma_g, \sigma_H$ given by Equations (6) and (7). $\square$

We next prove Lemma 3.7, establishing that all iterates stay inside a bounded region in which the covariance bounds are valid. The analysis utilizes a "dissipativity" property [Hal10]), which says that the iterate aligns with the direction of the gradient when the iterate's norm is large. For the gradient step, the gradient will reduce the norm of the iterate. For the negative curvature step, dissipativity property provides a lower bound on the gradient when the iterate's norm is large, but we show that Algorithm A.1 only takes a negative curvature step when this lower bound is violated, therefore the iterate's norm must be small and negative curvature steps with a fixed and small stepsize cannot increase the iterate's norm by too much.

*Proof of Lemma 3.7.* Recall Algorithm A.1 consists of inexact gradient descent steps of size $1/L_g = \frac{1}{16\Gamma}$ and randomized inexact negative curvature steps of size

$$\frac{2\epsilon_H}{L_H} = \frac{\sigma_r^\star}{36\Gamma} \leq \frac{\Gamma^{1/2}}{36}. \tag{25}$$

We proceed to use induction to prove the following for Algorithm A.1:

1. Suppose at step $\tau$ we run the negative curvature step to update the iterate from $U_\tau$ to $U_{\tau+1}$, then $\|U_\tau\|_{\mathrm{op}} \leq \frac{1}{2}\Gamma^{1/2}$

2. $\|U_t\|_{\mathrm{op}} \leq \Gamma^{1/2}$ for all $t \geq 0$.

First we consider the inexact gradient steps. We denote the gradient inexactness by $e_t = \nabla \bar{f}(U_t) - \widetilde{g}_t$. Recall that

$$\|e_t\|_F \leq 4rC_{est}\sigma_1^\star \|U_t\|_{op} \sqrt{\epsilon}$$

according to Equation (23) in Lemma C.5 whenever $\|U_t\|_{op} \leq \Gamma^{1/2}$, which is true according to the induction hypothesis. Therefore, we have

$$
\begin{aligned}
\|U_{t+1}\|_{op} &= \left\| U_t - \frac{1}{16\Gamma}\widetilde{g}_t \right\|_{op} = \left\| U_t - \frac{1}{16\Gamma}(2(U_tU_t - M^*)U_t - e_t) \right\|_{op} \\
&\leq \left\| U_t - \frac{1}{8\Gamma}U_tU_t^\top U_t \right\|_{op} + \frac{1}{8\Gamma}\left\| M^*U_t + \frac{1}{2}e_t \right\|_{op} \\
&\leq \max_i\{\sigma_i(U_t) - \frac{1}{8\Gamma}\sigma_i^3(U_t)\} + \frac{1}{8\Gamma}(\sigma_1^\star + 2rC_{est}\sigma_1^\star\sqrt{\epsilon})\|U_t\|_{op}
\end{aligned}
$$

Since the function $t \mapsto t - \frac{1}{8\Gamma}t^3$ is increasing in $[0, \sqrt{\frac{8\Gamma}{3}}]$ and $\|U_t\|_{op} \leq \Gamma^{1/2} \leq \sqrt{\frac{8\Gamma}{3}}$ by induction hypothesis, the maximum is taken when $i = 1$, hence

$$
\begin{aligned}
\|U_{t+1}\|_{op} &\leq \|U_t\|_{op} - \frac{1}{8\Gamma}\|U_t\|_{op}^3 + \frac{1}{8\Gamma}(\sigma_1^\star + 2rC_{est}\sigma_1^\star\sqrt{\epsilon})\|U_t\|_{op} \\
&\leq \|U_t\|_{op} - \frac{1}{8\Gamma}(\|U_t\|_{op}^2 - \sigma_1^\star - 2rC_{est}\sigma_1^\star\sqrt{\epsilon})\|U_t\|_{op} \\
&\leq \|U_t\|_{op} - \frac{1}{8\Gamma}(\|U_t\|_{op}^2 - 2\sigma_1^\star)\|U_t\|_{op}, \qquad (26)
\end{aligned}
$$

where the last inequality uses $\epsilon = O\left(\frac{1}{\kappa^3 r^3}\right) = O\left(\frac{1}{r^2}\right)$. We split into two cases now:

**Case** $\|U_t\|_{op} > \frac{1}{2}\Gamma^{1/2}$. Recall that $\Gamma \geq 36\sigma_1^\star$, hence

$$
\begin{aligned}
\|U_{t+1}\|_{op} &\leq \|U_t\|_{op} - \frac{1}{8\Gamma}(\|U_t\|_{op}^2 - 2\sigma_1^\star)\|U_t\|_{op} \\
&\leq \|U_t\|_{op} - \frac{1}{8\Gamma}(\frac{\Gamma}{4} - 2\sigma_1^\star)\frac{\Gamma^{1/2}}{2} \\
&\leq \|U_t\|_{op} - \frac{1}{8\Gamma}(\frac{\Gamma}{4} - \frac{\Gamma}{18})\frac{\Gamma^{1/2}}{2} \\
&\leq \|U_t\|_{op} - \frac{1}{96}\Gamma^{1/2},
\end{aligned}
$$

where the second inequality is because $t \mapsto (t^2 - 2\sigma_1^\star)t$ is increasing on $\left[\sqrt{\frac{2\sigma_1^\star}{3}}, +\infty\right]$ and $\|U\|_{op} \geq \frac{1}{2}\Gamma^{1/2} \geq 3\sigma_1^\star$. This case captures the dissipativity condition satisfied by the matrix sensing problem, which says that the iterate aligns with the direction of the gradient when the iterate's norm is large, so descending along the gradient decreases the norm of the iterate.

**Case** $\|U_t\|_{op} \leq \frac{1}{2}\Gamma^{1/2}$. From Equation (26) we know that if $\|U_t\|_{op}^2 \geq 2\|M^*\|_{op}$, it always holds that $\|U_{t+1}\|_{op} \leq \|U_t\|_{op} \leq \frac{1}{2}\Gamma^{1/2}$. For $\|U_t\|_{op}^2 \leq 2\|M^*\|_{op}$, we have $\|U_t\|_{op} \leq \sqrt{2}\sigma_1^{\star 1/2}$, and therefore

$$
\begin{aligned}
\|U_{t+1}\|_{op} &\leq \|U_t\|_{op} - \frac{1}{8\Gamma}(\|U_t\|_{op}^2 - 2\sigma_1^\star)\|U_t\|_{op} \\
&\leq \|U_t\|_{op} + \frac{1}{8\Gamma}(\sqrt{2}\sigma_1^{\star 1/2} + \|U_t\|_{op})((\sqrt{2}\sigma_1^{\star 1/2} - \|U_t\|_{op}))\|U_t\|_{op} \\
&\leq \|U_t\|_{op} + \frac{2\sqrt{2}\sigma_1^{\star 1/2}}{8\Gamma}(\sqrt{2}\sigma_1^{\star 1/2} - \|U_t\|_{op})\sqrt{2}\sigma_1^{\star 1/2} \\
&= \|U_t\|_{op} + \frac{\sigma_1^\star}{2\Gamma}(\sqrt{2}\sigma_1^\star - \|U_t\|_{op}) \\
&\leq \sqrt{2}\sigma_1^{\star 1/2} \qquad \text{(because the above expression is increasing in } \|U_t\|_{op}) \\
&\leq \frac{\sqrt{2}}{6}\Gamma^{1/2} < \Gamma^{1/2},
\end{aligned}
$$

where in the second last inequality we use $\sigma_1^\star \leq \Gamma/36$.

In summary, for all $t \in \mathbb{N}$,

$$\|U_t\|_{\text{op}} > \frac{1}{2}\Gamma^{1/2} \implies \left\|U_t - \frac{1}{16\Gamma}\widetilde{g}_t\right\|_{\text{op}} \leq \|U_t\|_{\text{op}} - \frac{1}{96}\Gamma^{1/2} \tag{27}$$

$$\|U_t\|_{\text{op}} \leq \frac{1}{2}\Gamma^{1/2} \implies \left\|U_t - \frac{1}{16\Gamma}\widetilde{g}_t\right\|_{\text{op}} \leq \frac{1}{2}\Gamma^{1/2} \tag{28}$$

In particular, for gradient step regime $U_{t+1} := U_t - \frac{1}{16\Gamma}\widetilde{g}_t$, we know the iterate $U_{t+1}$ stays inside the region $\{U : \|U\|_{\text{op}}^2 \leq \Gamma\}$ provided that $U_t$ is inside the region.

We proceed to analyze negative curvature steps, which only happen if the inexact gradient is small $\|\widetilde{g}_t\|_F \leq \epsilon_g = \frac{1}{32}\sigma_r^{\star 3/2}$. Note that it follows from Equation (27) that

$$\|\widetilde{g}_t\|_F \leq \frac{1}{32}\sigma_r^{\star 3/2} \implies \|\widetilde{g}_t\|_{\text{op}} \leq \frac{1}{32}\sigma_r^{\star 3/2} \leq \frac{1}{32}\Gamma^{3/2}$$

$$\implies \left\|U_t - \frac{1}{16\Gamma}\widetilde{g}_t\right\|_{\text{op}} \geq \|U_t\|_{\text{op}} - \frac{1}{512}\Gamma^{1/2} > \|U_t\|_{\text{op}} - \frac{1}{96}\Gamma^{1/2}$$

$$\implies \|U_t\|_{\text{op}} \leq \frac{1}{2}\Gamma^{1/2}.$$

Therefore, if at step $\tau$ we run the inexact negative curvature step to update the iterate from $U_\tau$ to $U_{\tau+1}$, then $\|U_\tau\|_{\text{op}} \leq \frac{1}{2}\Gamma^{1/2}$. Recall in Equation (25) the negative curvature stepsize $\frac{2\epsilon_H}{L_H} \leq \frac{\Gamma^{1/2}}{36} < \frac{\Gamma^{1/2}}{2}$, so for the negative curvature update

$$U_{\tau+1} = U_\tau + \frac{2\epsilon_H}{L_H}\sigma_\tau \widetilde{p}_\tau$$

where $\sigma_\tau = \pm 1$ with probability $1/2$ and $p_\tau$ is a vector with $\|p_\tau\|_F = 1$, we have

$$\|U_{\tau+1}\|_{\text{op}} \leq \|U_\tau\|_{\text{op}} + \left\|\frac{2\epsilon_H}{L_H}\sigma_\tau \widetilde{p}_\tau\right\|_{\text{op}}$$

$$\leq \|U_\tau\|_{\text{op}} + \left\|\frac{2\epsilon_H}{L_H}\sigma_\tau \widetilde{p}_\tau\right\|_F < \frac{1}{2}\Gamma^{1/2} + \frac{1}{2}\Gamma^{1/2} = \Gamma^{1/2}.$$

Finally, the initialization satisfies $\|U_0\|_{\text{op}}^2 \leq \Gamma$ by assumption, so the induction is complete. $\qquad\square$

### C.3  Local Linear Convergence in Low Rank Matrix Sensing—Omitted Proofs

Recall that Theorem 3.5 gives us a $(\frac{1}{24}\sigma_r^{\star 3/2}, \frac{1}{3}\sigma_r^\star)$-approximate SOSP as the initialization for Algorithm 3.1. We now explain why approximate SOSP is useful for the local search in the low rank matrix sensing problems.

**Definition C.6** (Strict Saddle Property)**.** Function $f(\cdot)$ is a $(\epsilon_g, \epsilon_H, \zeta)$-strict saddle function if for any $U \in \mathbb{R}^{r \times d}$, at least one of following properties holds:

(a) $\|\nabla f(U)\| > \epsilon_g$.

(b) $\lambda_{\min}(\nabla^2 f(U)) < -\epsilon_H$.

(c) $U$ is $\zeta$-close to $\mathcal{X}^\star$ — the set of local minima; namely, $\text{dist}(U, \mathcal{X}^\star) \leq \zeta$.

Approximate SOSPs are close to the set of local minima for strict saddle functions:

**Proposition C.7.** *Let $f$ be a $(\epsilon_g, \epsilon_H, \zeta)$-strict saddle function and $\mathcal{X}^\star$ be the set of its local minima. If $U_{SOSP}$ is a $(\epsilon_g, \epsilon_H)$-approximate SOSP of $f$, then $\text{dist}(U_{SOSP}, \mathcal{X}^\star) \leq \zeta$.*

*Proof.* Since $U_{SOSP}$ is a $(\epsilon_g, \epsilon_H)$-approximate SOSP of $f$, we have $\|\nabla f(U)\| \leq \epsilon_g$ and $\lambda_{\min}(\nabla^2 f(U)) \geq -\epsilon_H$, i.e., (a) and (b) in Definition C.6 fail to hold. Therefore, (c) holds, i.e., $\text{dist}(U, \mathcal{X}^\star) \leq \zeta$. $\qquad\square$

Some regularity conditions only hold in a local neighborhood of $\mathcal{X}^\star$. We focus on the following definition.

**Definition C.8** (Local Regularity Condition)**.** In a $\zeta$-neighborhood of the set of local minima $\mathcal{X}^\star$ (that is, $\text{dist}(U, \mathcal{X}^\star) \leq \zeta$), the function $f(\cdot)$ satisfies an $(\alpha, \beta)$-regularity condition if for any $U$ in this neighborhood:

$$\langle \nabla f(U), U - \mathcal{P}_{\mathcal{X}^\star}(U) \rangle \geq \frac{\alpha}{2} \|U - \mathcal{P}_{\mathcal{X}^\star}(U)\|^2 + \frac{1}{2\beta} \|\nabla f(U)\|^2. \tag{29}$$

Local regularity condition is a weaker condition than strong convexity, but both conditions would allow a first-order algorithm to obtain local linear convergence.

Recall that $\bar{f}(U) = \frac{1}{2} \|UU^\top - M^*\|_F^2$. The global minima of $\bar{f}$ solve the so-called symmetric low rank matrix factorization problem and the corresponding function value is zero. Let $TDT^\top$ be the singular value decomposition of the real symmetric matrix $M^*$. We observe $U^* = TD^{1/2}$ is a global optimum of $\bar{f}$.

The function $\bar{f}$ satisfies the local regularity condition and the strict saddle property, and all of its local minima are global minima [Jin+17].

**Fact C.9** ([Jin+17], Lemma 7)**.** *For $\bar{f}$ defined in (4), all local minima are global minima. The set of global minima is characterized by $\mathcal{X}^\star = \{U^* R | RR^\top = R^\top R = I\}$. Furthermore, $\bar{f}(U)$ satisfies:*

1. *$(\frac{1}{24}(\sigma_r^\star)^{3/2}, \frac{1}{3}\sigma_r^\star, \frac{1}{3}(\sigma_r^\star)^{1/2})$-strict saddle property, and*

2. *$(\frac{2}{3}\sigma_r^\star, 10\sigma_1^\star)$-regularity condition in the $\frac{1}{3}(\sigma_r^\star)^{1/2}$-neighborhood of $\mathcal{X}^\star$ in Frobenius norm.*

Similar to Corollary C.5, we need a theorem to say that all calls to RobustMeanEstimation (Algorithm A.2) are successful. For the local search (Algorithm 3.1), only inexact gradient oracles are required. Since gradients have dimension $dr$, the sample complexity needed for all robust estimates of gradient to be accurate with high probability is $\widetilde{O}\left(\frac{dr}{\epsilon}\right)$.

**Corollary C.10.** *Consider the noiseless setting as in Theorem I.7 with $\sigma = 0$. There exists a sample size $n = \widetilde{O}\left(\frac{dr}{\epsilon}\right)$ such that with $n$ samples, all subroutine calls to RobustMeanEstimation (Algorithm A.2) to estimate gradients $\nabla \bar{f}(U_t)$ succeed with high probability. In light of Equation (6), this means that there exists a large enough constant $C_{est}$ such that with high probability which is suppresed by $\widetilde{O}(\cdot)$, for all iterations $t$, we have*

$$\left\|\widetilde{g}_t - \nabla \bar{f}(U_t)\right\|_F \leq 2C_{est} \left\|U_t U_t^\top - M^*\right\|_F \|U_t\|_{\text{op}}. \tag{30}$$

*Proof.* The proof follows exactly the same line of argument as Corollary C.5 and Theorem B.3, which discusses the sample complexity required for robust mean estimation of both gradients and Hessians to be successful. Since the Hessian has dimension $d^2 r^2$, the sample complexity in Corollary C.5 is dominated by $\widetilde{\Omega}(d^2 r^2/\epsilon)$. Here the gradient has dimension $dr$, so we have the required sample complexity $\widetilde{\Omega}(dr/\epsilon)$. $\square$

*Proof of Theorem 3.8.* Let $e_t = \widetilde{g}_t - \nabla \bar{f}(U_t)$ and we rewrite the inexact gradient descent update as

$$U_{t+1} = U_t - \eta(\nabla \bar{f}(U_t) + e_t).$$

It follows from Corollary C.10 that, with high probability and for some large constant $C_{est}$, all calls to **RobustMeanEstimation** are successful, i.e., for all iterations $t$,

$$\|e_t\|_F \leq 2C_{est} \Gamma^{1/2} \left\|U_t U_t^\top - M^*\right\|_F \sqrt{\epsilon}. \tag{31}$$

We condition on this event for the rest of the analysis.

It is straightforward to check that $\frac{1}{3}\sigma_r^{\star 1/2}$-neighborhood of $\mathcal{X}^\star$ in Frobenius norm is inside the region $\{U : \|U\|_{\text{op}}^2 \leq \Gamma\}$. By strict saddle property in Fact C.9, we know from Proposition C.7 that $U_0 = U_{SOSP}$ is $\frac{1}{3}\sigma_r^{\star 1/2}$-close to $\mathcal{X}^\star$ in Frobenius norm. We assume for the sake of induction that $U_t$ is $\frac{1}{3}\sigma_r^{\star 1/2}$-close to $\mathcal{X}^\star$.

Let $\mathcal{P}_{\mathcal{X}^\star}(U)$ be the Frobenius projection of $U \in \mathbb{R}^{d \times r}$ onto $\mathcal{X}^\star$. By Fact C.9, for a given $U \in \mathbb{R}^{d \times r}$, there exists a rotation $R_U$ so that $\mathcal{P}_{\mathcal{X}^\star}(U) = U^\star R_U$. Therefore, $M^\star = U^\star U^{\star\top} = U^\star R_U R_U^\top U^{\star\top} = \mathcal{P}_{\mathcal{X}^\star}(U)\mathcal{P}_{\mathcal{X}^\star}(U)^\top$. We have

$$UU^\top - M^\star = UU^\top - U\mathcal{P}_{\mathcal{X}^\star}(U)^\top + U\mathcal{P}_{\mathcal{X}^\star}(U)^\top - \mathcal{P}_{\mathcal{X}^\star}(U)\mathcal{P}_{\mathcal{X}^\star}(U)^\top$$
$$= U(U - \mathcal{P}_{\mathcal{X}^\star}(U))^\top + (U - \mathcal{P}_{\mathcal{X}^\star}(U))\mathcal{P}_{\mathcal{X}^\star}(U)^\top.$$

Since $\|U_t\|_{\mathrm{op}} \le \Gamma^{1/2}$ and $\|\mathcal{P}_{\mathcal{X}^\star}(U)\|_{\mathrm{op}} \le \Gamma^{1/2}$, it follows from (31) that

$$\|e_t\|_F \le 2C_{est}\Gamma^{1/2}\sqrt{\epsilon} \left\|U_t U_t^\top - M^\star\right\|_F \le 4C_{est}\Gamma\sqrt{\epsilon} \left\|U_t - \mathcal{P}_{\mathcal{X}^\star}(U_t)\right\|_F.$$

We proceed to derive the exponential decrease of the distance to global minima.

$$\|U_{t+1} - \mathcal{P}_{\mathcal{X}^\star}(U_{t+1})\|_F^2$$
$$\le \|U_{t+1} - \mathcal{P}_{\mathcal{X}^\star}(U_t)\|_F^2$$
$$= \left\|U_t - \eta(\nabla\bar{f}(U_t) + e_t) - \mathcal{P}_{\mathcal{X}^\star}(U_t)\right\|_F^2$$
$$= \|U_t - \mathcal{P}_{\mathcal{X}^\star}(U_t)\|_F^2 + \eta^2\left\|\nabla\bar{f}(U_t) + e_t\right\|_F^2 - 2\eta\langle\nabla\bar{f}(U_t) + e_t, U_t - \mathcal{P}_{\mathcal{X}^\star}(U_t)\rangle$$
$$\le (1 - \eta\alpha)\|U_t - \mathcal{P}_{\mathcal{X}^\star}(U_t)\|_F^2 + \eta^2\left\|\nabla\bar{f}(U_t) + e_t\right\|_F^2 - \frac{\eta}{\beta}\left\|\nabla\bar{f}(U_t)\right\|_F^2 - 2\eta\langle e_t, U_t - \mathcal{P}_{\mathcal{X}^\star}(U_t)\rangle,$$

where the last inequality comes from the regularity condition in Fact C.9 with $\alpha = \frac{2}{3}\sigma_r^\star, \beta = 10\sigma_1^\star$. The cross term can be bounded by

$$-2\eta\langle e_t, U_t - \mathcal{P}_{\mathcal{X}^\star}(U_t)\rangle \le 2\eta\|e_t\|_F\|U_t - \mathcal{P}_{\mathcal{X}^\star}(U_t)\|_F \le 8C_{est}\eta\Gamma\sqrt{\epsilon}\|U_t - \mathcal{P}_{\mathcal{X}^\star}(U_t)\|_F^2,$$

where the last inequality comes from Equation 31.

By Young's inequality,

$$\eta^2\left\|\nabla\bar{f}(U_t) + e_t\right\|_F^2 \le 2\eta^2\left\|\nabla\bar{f}(U_t)\right\|_F^2 + 2\eta^2\|e_t\|_F^2$$
$$\le 2\eta^2\left\|\nabla\bar{f}(U_t)\right\|_F^2 + 32\eta^2 C_{est}^2\Gamma^2\epsilon\|U_t - \mathcal{P}_{\mathcal{X}^\star}(U_t)\|_F^2.$$

Recall that $\Gamma \ge 36\sigma_1^\star$. Choosing $\eta = \frac{1}{\Gamma} \le \frac{1}{36\sigma_1^\star} < \frac{1}{20\sigma_1^\star} = \frac{1}{2\beta}$, it follows that

$$\|U_{t+1} - \mathcal{P}_{\mathcal{X}^\star}(U_{t+1})\|_F^2$$
$$\le (1 - \eta\alpha)\|U_t - \mathcal{P}_{\mathcal{X}^\star}(U_t)\|_F^2 + \eta^2\left\|\nabla\bar{f}(U_t) + e_t\right\|_F^2 - \frac{\eta}{\beta}\left\|\nabla\bar{f}(U_t)\right\|_F^2 - 2\eta\langle e_t, U_t - \mathcal{P}_{\mathcal{X}^\star}(U_t)\rangle$$
$$\le (1 - \eta\alpha + 8C_{est}\eta\Gamma\sqrt{\epsilon} + 32\eta^2 C_{est}^2\Gamma^2\epsilon)\|U_t - \mathcal{P}_{\mathcal{X}^\star}(U_t)\|_F^2 + (2\eta^2 - \frac{\eta}{\beta})\left\|\nabla\bar{f}(U_t)\right\|_F^2$$
$$\le \left(1 - \frac{\sigma_r^\star}{30\Gamma} + \frac{2}{5}C_{est}\sqrt{\epsilon} + 32(C_{est}/20)^2\epsilon\right)\|U_t - \mathcal{P}_{\mathcal{X}^\star}(U_t)\|_F^2$$
$$\le \left(1 - \left(O\left(\frac{1}{\kappa}\right) - O(\sqrt{\epsilon})\right)\right)\|U_t - \mathcal{P}_{\mathcal{X}^\star}(U_t)\|_F^2. \tag{32}$$

One consequence of this calculation is $\|U_{t+1} - \mathcal{P}_{\mathcal{X}^\star}(U_{t+1})\|_F \le \|U_t - \mathcal{P}_{\mathcal{X}^\star}(U_t)\|_F$, so $U_{t+1}$ is also $\frac{1}{3}\sigma_r^{\star 1/2}$-close to $\mathcal{X}^\star$ in Frobenius norm, completing the induction. The other consequence is local linear convergence of $U_t$ to $\mathcal{X}^\star$ with rate $\left(1 - O\left(\frac{1}{\kappa}\right)\right)$, assuming $\epsilon\kappa^2$ is sufficiently small. Since the initial distance is bounded by $O(\sigma_r^{\star 1/2})$, converging to a point that is $\iota$-close to $\mathcal{X}^\star$ in Frobenius norm requires the following number of iterations:

$$O\left(\frac{\log(\iota/\sigma_r^{\star 1/2})}{\log(1 - O(1/\kappa))}\right) = O\left(\kappa\log\left(\frac{\sigma_r^\star}{\iota}\right)\right). \qquad \square$$

## C.4  Low Rank Matrix Sensing with Noise

In Section 3.1, we focused on the case that $\sigma = 0$ as in Theorem 3.2. Now we consider the case that $\sigma \neq 0$ and prove Theorem 3.3.

Recall that $y_i \sim \mathcal{N}(\langle A_i, M^* \rangle, \sigma^2)$ in Definition 1.6. Since $\langle A_i, M^* \rangle$ follows Gaussian distribution with mean 0 and variance $\|M^*\|_F^2$, we optionally assume $\sigma = O(\|M^*\|_F) = O(r\Gamma)$ to keep the signal-to-noise ratio in constant order. For the ease of presentation, we make the following assumption:

**Assumption C.11.** *Assume $\sigma \leq r\Gamma$.*

We present different algorithms depending on whether Assumption C.11 holds.

As in the noiseless case, we define $f_i(U) = \frac{1}{2} \left( \langle UU^\top, A_i \rangle - y_i \right)^2$ and

$$\bar{f}(U) = \mathop{\mathbb{E}}_{(A_i, y_i) \sim \mathcal{G}_\sigma} f_i(U) = \frac{1}{2} \left\| UU^\top - M^* \right\|_F^2 + \frac{1}{2}\sigma^2.$$

Note that the $\frac{1}{2}\sigma^2$ term in $\bar{f}(U)$ has no effect on its minimum, gradients, or Hessians, and Fact C.9 and 3.4 still apply in verbatim.

We prove the following result when Assumption C.11 holds:

**Theorem C.12.** *Consider the same setting as in Theorem 1.7 with $0 \neq \sigma \leq r\Gamma$ (Assumption C.11 holds). There exists a sample size $n = \widetilde{O}(d^2 r^2/\epsilon)$ such that with high probability, there exists an algorithm that outputs a solution $\widehat{M}$ in $\widetilde{O}(r^2\kappa^3)$ calls to robust mean estimation routine A.2, with error $\|\widehat{M} - M^*\|_F = O(\kappa\sigma\sqrt{\epsilon})$.*

*Proof.* We compute the gradient and Hessian of $f_i$:

$$\nabla f_i(U) = \left( \langle UU^\top - M^*, A_i \rangle - \zeta_i \right) (A_i + A_i^\top)U$$
$$H_i(U) = \left( \langle UU^\top - M^*, A_i \rangle - \zeta_i \right) I_r \otimes B_i + \text{vec}\left(B_i U\right) \text{vec}(B_i U)^\top.$$

We also need to bound the covariance of sample gradients and sample Hessians in the bounded region $\{U \in \mathbb{R}^{d \times r} : \|U\|_{\text{op}}^2 \leq \Gamma\}$, similar to Equations (6) and (7).

$$\left\| \text{Cov}(\text{vec}(\nabla f_i(U))) \right\|_{\text{op}} \leq 4 \left( \left\| UU^\top - M^* \right\|_F^2 + \sigma^2 \right) \|U\|_{\text{op}}^2 = O\!\left(r^2\Gamma^3\right). \tag{33}$$

$$\left\| \text{Cov}(\text{vec}(H_i)) \right\|_{\text{op}} \leq 16r \left( \left\| UU^\top - M^* \right\|_F^2 + \sigma^2 \right) + 128 \|U\|_{\text{op}}^4 = O\!\left(r^3\Gamma^2\right). \tag{34}$$

Since global convergence analysis in Section 3.1.2 only requires $\left\| \text{Cov}(\text{vec}(\nabla f_i(U))) \right\|_{\text{op}} = O\!\left(r^2\Gamma^3\right)$ and $\left\| \text{Cov}(\text{vec}(H_i)) \right\|_{\text{op}} = O\!\left(r^3\Gamma^2\right)$, Theorem 3.5 also holds in the noisy setting (up to a constant factor). This means that we could obtain a $(\frac{1}{24}\sigma_r^{\star 3/2}, \frac{1}{3}\sigma_r^\star)$-approximate SOSP of $\bar{f}$, which is in $\frac{1}{3}\sigma_r^{\star 1/2}$-neighborhood of $\mathcal{X}^\star$ in Frobenius norm.

We now focus on the local convergence analysis. We use the same algorithm (Algorithm 3.1), which uses RobustMeanEstimation subroutine in each iteration $t$ to obtain an inexact gradient $\widetilde{g}_t$ for $\bar{f}(U)$. We rewrite the inexact gradient descent update as

$$U_{t+1} = U_t - \eta(\nabla \bar{f}(U_t) + e_t).$$

We condition the remaining analysis on the event that all calls to **RobustMeanEstimation** are successful as in Corollary C.10, which happens with high probability and implies

$$\|e_t\|_F \leq 2C_{est} \left( \left\| U_t U_t^\top - M^* \right\|_F + \sigma \right) \|U_t\|_{\text{op}} \sqrt{\epsilon}$$
$$\leq 2C_{est} \left( \left\| U_t U_t^\top - M^* \right\|_F + \sigma \right) \Gamma^{1/2}\sqrt{\epsilon}.$$

It is straightforward to check that $\frac{1}{3}\sigma_r^{\star 1/2}$-neighborhood of $\mathcal{X}^\star$ in Frobenius norm is inside the region $\{U : \|U\|_{\text{op}}^2 \leq \Gamma\}$. By strict saddle property in Fact C.9, we know $U_0 = U_{SOSP}$ is $\frac{1}{3}\sigma_r^{\star 1/2}$-close

to $\mathcal{X}^\star$ in Frobenius norm. We assume for the sake of induction that $U_t$ is $\frac{1}{3}\sigma_r^{\star 1/2}$-close to $\mathcal{X}^\star$. We will show that $\|U_{t+1} - \mathcal{P}_{\mathcal{X}^\star}(U_{t+1})\|_F$ either shrinks or is at the order of $O\left(\frac{\sigma\epsilon}{\Gamma^{1/2}}\right)$, which is also $\frac{1}{3}\sigma_r^{\star 1/2}$-close to $\mathcal{X}^\star$ in Frobenius norm for sufficiently small $\epsilon = O\left(\frac{1}{r\kappa}\right)$.

As before, we define $\mathcal{P}_{\mathcal{X}^\star}(\cdot)$ to be the projection onto $\mathcal{X}^\star$ and bound the Frobenius distance to $\mathcal{X}^\star$ as follows:

$$\|U_{t+1} - \mathcal{P}_{\mathcal{X}^\star}(U_{t+1})\|_F^2$$
$$\leq \|U_{t+1} - \mathcal{P}_{\mathcal{X}^\star}(U_t)\|_F^2$$
$$= \|U_t - \eta(\nabla \bar{f}(U_t) + e_t) - \mathcal{P}_{\mathcal{X}^\star}(U_t)\|_F^2$$
$$= \|U_t - \mathcal{P}_{\mathcal{X}^\star}(U_t)\|_F^2 + \eta^2 \|\nabla \bar{f}(U_t) + e_t\|_F^2 - 2\eta \langle \nabla \bar{f}(U_t) + e_t, U_t - \mathcal{P}_{\mathcal{X}^\star}(U_t) \rangle$$
$$\leq (1 - \eta\alpha) \|U_t - \mathcal{P}_{\mathcal{X}^\star}(U_t)\|_F^2 + \eta^2 \|\nabla \bar{f}(U_t) + e_t\|_F^2 - \frac{\eta}{\beta} \|\nabla \bar{f}(U_t)\|_F^2 - 2\eta \langle e_t, U_t - \mathcal{P}_{\mathcal{X}^\star}(U_t) \rangle,$$

where the last inequality comes from the regularity condition in Fact C.9 with $\alpha = \frac{2}{3}\sigma_r^\star, \beta = 10\sigma_1^\star$.

Using Cauchy-Schwarz inequality to bound $\|\nabla \bar{f}(U_t) + e_t\|_F^2$ and $\langle e_t, U_t - \mathcal{P}_{\mathcal{X}^\star}(U_t) \rangle$ and choosing $\eta = \frac{1}{20\Gamma} \leq \frac{1}{20\sigma_1^\star} = \frac{1}{2\beta}$ to kill the $\|\nabla \bar{f}(U_t)\|_F^2$ term, we obtain

$$\|U_{t+1} - \mathcal{P}_{\mathcal{X}^\star}(U_{t+1})\|_F^2$$
$$\leq (1 - \eta\alpha) \|U_t - \mathcal{P}_{\mathcal{X}^\star}(U_t)\|_F^2 + 2\eta^2 \|\nabla \bar{f}(U_t)\|_F^2 + 2\|e_t\|_F^2 - \frac{\eta}{\beta} \|\nabla \bar{f}(U_t)\|_F^2 - 2\eta \langle e_t, U_t - \mathcal{P}_{\mathcal{X}^\star}(U_t) \rangle$$
$$\leq (1 - \eta\alpha) \|U_t - \mathcal{P}_{\mathcal{X}^\star}(U_t)\|_F^2 + 2\eta^2 \|e_t\|_F^2 - 2\eta \langle e_t, U_t - \mathcal{P}_{\mathcal{X}^\star}(U_t) \rangle + (2\eta^2 - \frac{\eta}{\beta}) \|\nabla \bar{f}(U_t)\|_F^2$$
$$\leq (1 - \eta\alpha) \|U_t - \mathcal{P}_{\mathcal{X}^\star}(U_t)\|_F^2 + 2\eta^2 \|e_t\|_F^2 + 2\eta \|e_t\|_F \|U_t - \mathcal{P}_{\mathcal{X}^\star}(U_t)\|_F.$$

Now we consider the following two possible cases:

**Case 1:** $\|U_t U_t^\top - M^*\|_F > \sigma$. Then
$$\|e_t\|_F \leq 2C_{est} \left(\|U_t U_t^\top - M^*\|_F + \sigma\right) \Gamma^{1/2}\sqrt{\epsilon}$$
$$< 4C_{est} \|U_t U_t^\top - M^*\|_F \Gamma^{1/2}\sqrt{\epsilon}.$$

And it follows similar to (32) that

$$\|U_{t+1} - \mathcal{P}_{\mathcal{X}^\star}(U_{t+1})\|_F^2$$
$$\leq (1 - \eta\alpha + 8C_{est}\eta\Gamma\sqrt{\epsilon} + 32\eta^2 C_{est}^2 \Gamma^2 \epsilon) \|U_t - \mathcal{P}_{\mathcal{X}^\star}(U_t)\|_F^2$$
$$\leq \left(1 - \frac{\sigma_r^\star}{30\Gamma} + \frac{2}{5}C_{est}\sqrt{\epsilon} + 32(C_{est}/20)^2\epsilon\right) \|U_t - \mathcal{P}_{\mathcal{X}^\star}(U_t)\|_F^2$$
$$\leq \left(1 - \left(O\left(\frac{1}{\kappa}\right) - O(\sqrt{\epsilon})\right)\right) \|U_t - \mathcal{P}_{\mathcal{X}^\star}(U_t)\|_F^2.$$

We conclude that the distance to $\mathcal{X}^\star$ decreases geometrically in this regime until $\|U_t U_t^\top - M^*\|_F \leq \sigma$, which takes a total of $O\left(\kappa \log\left(\frac{\sigma_r^\star}{\sigma}\right)\right)$ iterations, which is very few if $\sigma = \Theta(r\Gamma)$.

**Case 2:** $\|U_t U_t^\top - M^*\|_F \leq \sigma$. Then $\|e_t\|_F \leq 4C_{est}\sigma\Gamma^{1/2}\sqrt{\epsilon}$. It follows that

$$\|U_{t+1} - \mathcal{P}_{\mathcal{X}^\star}(U_{t+1})\|_F^2$$
$$\leq (1 - \eta\alpha) \|U_t - \mathcal{P}_{\mathcal{X}^\star}(U_t)\|_F^2 + 2\eta^2 \|e_t\|_F^2 + 2\eta \|e_t\|_F \|U_t - \mathcal{P}_{\mathcal{X}^\star}(U_t)\|_F$$
$$\leq (1 - \eta\alpha) \|U_t - \mathcal{P}_{\mathcal{X}^\star}(U_t)\|_F^2 + 32\eta^2 C_{est}^2 \sigma^2 \Gamma\epsilon + 8\eta C_{est}\sigma\Gamma^{1/2}\sqrt{\epsilon} \|U_t - \mathcal{P}_{\mathcal{X}^\star}(U_t)\|_F.$$

Write $\mathcal{C} = 4\sqrt{2}\eta C_{est}\sigma\Gamma^{1/2}\sqrt{\epsilon}, \mathcal{B}_t = \|U_{t+1} - \mathcal{P}_{\mathcal{X}^\star}(U_{t+1})\|_F / \mathcal{C}$, we have

$$\mathcal{B}_{t+1} \leq \sqrt{(1 - \eta\alpha)\mathcal{B}_t^2 + \mathcal{B}_t + 1}.$$

It is straightforward to compute that the function $b \mapsto \sqrt{(1 - \eta\alpha)b^2 + b + 1}$ has Lipschitz constant

$$\rho = \begin{cases} \frac{1}{2} & \text{if } 0 < 1 - \eta\alpha \leq \frac{1}{4} \\ \sqrt{1 - \eta\alpha} & \text{if } \frac{1}{4} \leq 1 - \eta\alpha < 1. \end{cases}$$

Since $\rho < 1$, Banach fixed-point theorem implies that $\mathcal{B}_t$ converges to some $\mathcal{B}^*$ with rate

$$|\mathcal{B}_{t+1} - \mathcal{B}^*| \leq \rho |\mathcal{B}_t - \mathcal{B}^*|.$$

We can compute an upper bound on the fixed point $\mathcal{B}^*$ by solving

$$\mathcal{B}^* = \sqrt{(1 - \eta\alpha)\mathcal{B}^{*2} + \mathcal{B}^* + 1},$$

which gives $\mathcal{B}^* = O\left(\frac{1}{\eta\alpha}\right) = O(\kappa)$.

Denote the first iteration of Case 2 regime ($\|U_t U_t^\top - M^*\|_F \leq \sigma$) by $t_0$. By [GJZ17, Lemma 6], we have $\|U_{t_0} - \mathcal{P}_{\mathcal{X}^*} U_{t_0}\|_F \leq \sigma\sqrt{\frac{2}{\sigma_r^*}}$ and therefore $\mathcal{B}_{t_0} = \sqrt{\kappa/\epsilon}$. Therefore, Case 2 takes a total of $O(\kappa \log 1/\epsilon)$ iterations to reach the error

$$\|U_t - \mathcal{P}_{\mathcal{X}^*}(U_t)\|_F \leq O\left(\kappa \Gamma^{-1/2} \sigma\sqrt{\epsilon}\right).$$

This translates to the error in the matrix space

$$\left\|U_t U_t^\top - M^*\right\|_F \leq \Gamma^{1/2} \left\|U_t - \mathcal{P}_{\mathcal{X}^*}(U_t)\right\|_F = O\left(\kappa\sigma\sqrt{\epsilon}\right).$$

$\square$

Assumption C.11 might fail to hold; a notable example is our construction in Section 4, where $\|M^*\|_F = O(\epsilon^{1/2})$ but $\sigma = O(1)$. In this case, we consider a simple spectral method, relying on the observation that with clean samples $\mathbb{E}_{(A_i, y_i) \in \mathcal{G}_\sigma}[y_i A_i] = M^*$. So we run RobustMeanEstimation on $\{y_i A_i\}_{i=1}^n$ to get $\widetilde{M}$ with its singular decomposition $\widetilde{M} = \sum_{i=1}^n s_i u_i v_i^\top$, where singular values $s_1 \geq s_2 \geq \ldots \geq s_n$ are descending. We then form a new matrix $\widehat{M}$ from the $r$ leading singular vectors, i.e., $\widehat{M} = \sum_{i=1}^r s_i u_i v_i^\top$.

This simple algorithm has the following guarantee:

**Theorem C.13.** *Consider the same setting as in Theorem I.7 with $\sigma > r\Gamma$, i.e., Assumption C.11 fails. There exists a sample size $n = \widetilde{O}(d^2/\epsilon)$ such that with high probability, there exists an algorithm that outputs a solution $\widehat{M}$ in one call to robust mean estimation routine A.2, with error $\|\widehat{M} - M^*\|_F = O(\sigma\sqrt{\epsilon})$.*

*Proof.* We compute the mean and variance of $\langle y_i A_i, Z \rangle$ for some unit Frobenius norm $Z$.

$$\mathbb{E}\langle y_i A_i, Z \rangle = \mathbb{E}\langle A_i, M^* \rangle \langle A_i, Z \rangle + \mathbb{E}\zeta\langle A_i, Z \rangle$$
$$= \mathbb{E}\langle A_i, M^* \rangle \langle A_i, Z \rangle = \langle M^*, Z \rangle.$$

Hence $\mathbb{E}[y_i A_i] = M^*$.

$$\text{Var}\left(\langle A_i, M^* \rangle \langle A_i, Z \rangle + \zeta\langle A_i, Z \rangle\right)$$
$$\leq 2\text{Var}\langle A_i, M^* \rangle \langle A_i, Z \rangle + 2\text{Var}(\zeta\langle A_i, Z \rangle)$$
$$\leq 4\|M^*\|_F^2 + 4\sigma^2 \leq 8\sigma^2,$$

where the last inequality is because $\|M^*\|_F \leq r\Gamma$. Hence running robust mean estimation algorithm on $\{y_i A_i\}$ outputs $\widetilde{M}$ with $\left\|\widetilde{M} - M^*\right\|_F = O(\sigma\sqrt{\epsilon})$.

Since $\widehat{M}$ is formed by the $r$ leading singular vectors of $\widetilde{M}$, it is the best rank $r$ approximation to $\widetilde{M}$ by Eckart–Young–Mirsky's theorem, i.e., $\left\|\widehat{M} - \widetilde{M}\right\|_F \leq \left\|M^* - \widetilde{M}\right\|_F = O(\sigma\sqrt{\epsilon})$. We conclude that $\left\|\widehat{M} - M^*\right\|_F \leq \left\|\widehat{M} - \widetilde{M}\right\|_F + \left\|\widetilde{M} - M^*\right\|_F = O(\sigma\sqrt{\epsilon})$ by triangle inequality. $\square$

# D   SQ Lower bound—Omitted Proofs

We consider a weaker corruption model, known as Huber's $\epsilon$-contamination model [Hub64], than Definition 1.1 in the sense that any corruptions in Huber's contamination model can be emulated by the adversary in Definition 1.1. We show a lower bound of the sample complexity in the presence of Huber's $\epsilon$-contamination, and the sample complexity under Definition 1.1 can therefore be no better.

**Assumption D.1** (Huber's Contamination Model). *Let $Q$ be the joint distribution of $(A, y)$ with $\mathrm{vec}(A) \sim \mathcal{N}(0, I_{d^2})$ and $y|A \sim \mathcal{N}(\langle M^*, uu^\top \rangle, \sigma^2)$. The input samples are contaminated according to Huber's $\epsilon$-contamination model in the following way: we observe samples from a mixture $Q'$ of the clean joint distribution $Q$ and an arbitrary noise distribution $N$, i.e., $Q' = (1 - \epsilon)Q + \epsilon N$.*

The adversary in the $\epsilon$-strong contamination model can emulate the Huber's adversary: on input clean samples, the $\epsilon$-strong contamination adversary removes a randomly chosen $\epsilon$-fraction of samples and replaces them with samples drawn i.i.d. from $N$.

Recall that SQ algorithms are a class of algorithms that, instead of access to samples, are allowed to query expectations of bounded functions of the distribution.

**Definition D.2** (SQ Algorithms and STAT Oracle [Kea98]). Let $\mathcal{G}$ be a distribution on $\mathbb{R}^{d^2+1}$. A Statistical Query (SQ) is a bounded function $q : \mathbb{R}^{d^2+1} \to [-1, 1]$. For $\tau > 0$, the $\mathsf{STAT}(\tau)$ oracle responds to the query $q$ with a value $v$ such that $|v - \mathbb{E}_{X \sim D}[q(X)]| \leq \tau$. An SQ algorithm is an algorithm whose objective is to learn some information about an unknown distribution $\mathcal{G}$ by making adaptive calls to the corresponding $\mathsf{STAT}(\tau)$ oracle.

A statistical query with accuracy $\tau$ can be implemented with error probability $\delta$ by taking $O(\log(1/\delta)/\tau^2)$ samples and evaluating the empirical mean of the query function $q(\cdot)$ evaluated at those samples [DK23, Chapter 8.2.1]. This bound is tight in general for a single query.

**Lemma D.3** (Clean marginal and conditional distribution). *Let $A \in \mathbb{R}^{d \times d}$ with $\mathrm{vec}(A) \sim \mathcal{N}(0, I_{d^2})$ and $y|A \sim \mathcal{N}(\langle uu^\top, M^* \rangle, \sigma^2)$. Then*

$$y \sim \mathcal{N}(0, \sigma^2 + \|u\|^4)$$

$$\mathrm{vec}(A)|y \sim \mathcal{N}\left( \frac{\mathrm{vec}(uu^\top)y}{\sigma^2 + \|u\|^4}, I_{d^2} - \frac{\mathrm{vec}(uu^\top)\mathrm{vec}(uu^\top)^\top}{\sigma^2 + \|u\|^4} \right).$$

*Proof.* Let $Q$ be the joint distribution of $(\mathrm{vec}(A), y)$. We can write down the covariance of $Q$:

$$\begin{bmatrix} I_{d^2} & \mathrm{vec}(uu^\top) \\ \mathrm{vec}(uu^\top)^\top & \sigma^2 + \|u\|^4 \end{bmatrix}.$$

The mean and variance of $\mathrm{vec}(A)|y$ follow from the conditional Gaussian formula.   $\square$

We will construct a family of contaminated joint distributions of $(A, y)$ that are hard to learn with SQ access. We first construct a family of clean distributions and then construct the contaminations.

**Lemma D.4** (Defining Clean Distributions). *Let $v$ be a unit vector and set $u = c_1^{1/2} \epsilon^{1/4} v$ for some small constant $c_1$. Let $\mathrm{vec}(A) \sim \mathcal{N}(0, I_{d^2})$, and choose $\sigma = 1 - c_1^2 \epsilon$. Then we have the marginal distribution $y \sim \mathcal{N}(0, 1)$ and*

$$\mathrm{vec}(A)|y \sim \mathcal{N}\left( c_1 \sqrt{\epsilon}\, \mathrm{vec}(vv^\top)y, I_{d^2} - c_1^2 \epsilon\, \mathrm{vec}(vv^\top)\mathrm{vec}(vv^\top)^\top \right).$$

Let $Q_v$ denote the joint distribution from the construction in Lemma D.4. The family of contaminated joint distributions that are hard to learn will be denoted by $Q'_v$, and we proceed to construct them. The technique for the construction follows a standard non-Gaussian component analysis argument, which starts from mixing a one-dimensional distribution to match moments with the standard Gaussian. We note that the conditional distribution $\mathrm{vec}(A)|y$ in all directions perpendicular to $\mathrm{vec}(vv^\top)$ is already standard Gaussian, so we proceed to consider the one-dimensional distribution along $\mathrm{vec}(vv^\top)$ direction, which is

$$\mathcal{N}\left( c_1 \sqrt{\epsilon} y, 1 - c_1^2 \epsilon \right).$$

Let $\mu(y) = c_1\sqrt{\epsilon}y$ be a shortand for the mean of this one dimensional distribution. We proceed to mix it with some noise distribution $B_\mu$. It is important to note here that the fraction of the noise depends on $\mu$ and thus on $y$—the larger $\mu$ is, the higher the noise level is needed.

**Definition D.5.** Let $P, Q$ be two distributions with probability density functions $P(x), Q(x)$; we obscure the line between a distribution and its density function. The $\chi^2$-divergence of $P$ and $Q$ is

$$\chi^2(P, Q) = \int \frac{(P(x) - Q(x))^2}{Q(x)} dx = \int \frac{P^2(x)}{Q(x)} dx - 1.$$

Let $N$ be a distribution whose support contains the supports of $P$ and $Q$. We define

$$\chi_N(P, Q) = \int \frac{P(x)Q(x)}{N(x)} dx.$$

Then $|\chi_N(P, Q) - 1| = \int \frac{(P(x) - N(x))(Q(x) - N(x))}{N(x)} dx$ defines an inner product of $P$ and $Q$.

**Lemma D.6** (One-Dimensional Moment Matching). *For any $\epsilon > 0, \mu \in \mathbb{R}$, there is a distribution $D_\mu$ such that $D_\mu$ agrees with the mean of $\mathcal{N}(0, 1)$ and*

$$D_\mu = (1 - \epsilon_\mu)\mathcal{N}(\mu, 1 - c_1^2\epsilon) + \epsilon_\mu B_\mu,$$

*for some distribution $B_\mu$ and $\epsilon_\mu$ satisfying*

1. *If $|\mu| < c_1^2\sqrt{\epsilon}$, then $\epsilon_\mu = \epsilon$ and $\chi^2(D_\mu, \mathcal{N}(0, 1)) = e^{O(1/\epsilon)}$.*

2. *If $|\mu| \geq c_1^2\sqrt{\epsilon}$, then we take $\epsilon_\mu$ such that $\epsilon_\mu/(1 - \epsilon_\mu) = \mu^2$, which simplies to*

$$\epsilon_\mu = \frac{\mu^2}{1 + \mu^2}, \tag{35}$$

*and $\chi^2(D_\mu, \mathcal{N}(0, 1)) = e^{O(\max(1/\mu^2, \mu^2))}$.*

*Proof.* It is straightforward to verify that $D_\mu = (1 - \epsilon_\mu)\mathcal{N}(\mu, 1 - c_1^2\epsilon) + \epsilon_\mu \mathcal{N}(a, b)$ has mean 0 and second moment 1, where $a$ and $b$ are defined by

$$a = -\frac{\mu(1 - \epsilon_\mu)}{\epsilon_\mu}, \ b = 1.$$

Facts D.11 and D.12 imply that if $\sigma_1 = O(1), \sigma_2 = O(1)$, then

$$\chi^2(\epsilon_\mu \mathcal{N}(\mu_1, \sigma_1^2) + (1 - \epsilon_\mu)\mathcal{N}(\mu_2, \sigma_2^2), \mathcal{N}(0, 1)) = e^{O(\mu_1^2 + \mu_2^2)}.$$

- **Case 1**: $|\mu| < c_1^2\sqrt{\epsilon}$. Then taking $\epsilon_\mu = \epsilon$ implies that $|a| \leq \mu/\epsilon < c_1/\sqrt{\epsilon}$, and $\chi^2(D_\mu, \mathcal{N}(0, 1)) = e^{O(1/\epsilon)}$.

- **Case 2**: $|\mu| \geq c_1^2\sqrt{\epsilon}$. Then taking $\epsilon_\mu/(1 - \epsilon_\mu) = \mu^2$ implies that $|a| = \frac{1}{|\mu|}$, and $\chi^2(D_\mu, \mathcal{N}(0, 1)) = e^{O(\max(1/\mu^2, \mu^2))}$. $\qquad\square$

We follow the non-Gaussian component analysis construction and define the conditional distribution of the corrupted $\text{vec}(A)|y$ to have a hidden direction $v$.

**Definition D.7** (High-Dimensional Hidden Direction Distribution [DK23, Definition 8.9]). For a one-dimensional distribution $D$ that admits density $D(x)$ and a unit vector $v \in \mathbb{R}^{d^2}$, the distribution $\mathbf{P}_v^A$ is defined to be the product distribution of $D(x)$ in the $v$-direction and standard normal distribution in the subspace perpendicular to $v$, i.e.,

$$\mathbf{P}_v^D(a) = \frac{1}{(2\pi)^{\frac{d^2-1}{2}}} D(v^\top a) \exp\left(-\frac{1}{2}\left\|a - (v^\top a)v\right\|^2\right).$$

**Lemma D.8** (Construction of Corrupted Conditional Distribution). *Define $P_{\mu(y),v} = \mathbf{P}_{\text{vec}(vv^\top)}^{D_{\mu(y)}}$. Then for all unit vectors $v, v' \in \mathbb{R}^d$, it holds that*

$$\left|\chi_{\mathcal{N}(0,I)}(P_{\mu(y),v}, P_{\mu(y),v'}) - 1\right| \leq (v^\top v')^4 \chi^2(D_{\mu(y)}, \mathcal{N}(0, 1)). \tag{36}$$

*Proof.* By Lemma 8.12 in the textbook [DK23], we have

$$\left| \chi_{\mathcal{N}(0,I)}(P_{\mu(y),v}, P_{\mu(y),v'}) - 1 \right| = \left| \chi_{\mathcal{N}(0,I)}(\mathbf{P}^{D_{\mu(y)}}_{\text{vec}(vv^\top)}, \mathbf{P}^{D_{\mu(y)}}_{\text{vec}(v'v'^\top)}) - 1 \right|$$
$$\leq (\text{vec}(vv^\top)^\top \text{vec}(v'v'^\top))^2 \chi^2(D_{\mu(y)}, \mathcal{N}(0,1))$$
$$= (v^\top v')^4 \chi^2(D_{\mu(y)}, \mathcal{N}(0,1)),$$

where the inequality is a direct application of [DK23, Lemma 8.12], and the last equality is because

$$\text{vec}(vv^\top)^\top \text{vec}(v'v'^\top) = \langle vv^\top, v'v'^\top \rangle = \text{tr}(vv^\top v'v'^\top) = v^\top v' \, \text{tr}(vv'^\top) = (v^\top v')^2.$$

$\square$

Now we define $Q'_v(A, y)$ to be a contaminated version of the joint distribution of $A$ and $y$ such that $\text{vec}(A)|y \sim P_{\mu(y),v}$ in the following lemma. Recall that, according to Lemma D.6, different level of noise $\epsilon_{\mu(y)}$ is needed for different $y$, and the implication of the following lemma is that the total noise is less than $\epsilon$. This is done via integration with respect to $y$.

**Lemma D.9** (Construction of Corrupted Joint Distribution: Controlling Large $\epsilon_{\mu(y)}$). *Recall $\mu(y) = c_1\sqrt{\epsilon}y$ and $Q_v$ is the joint distribution of $A$ and $y$ for clean samples. Define $Q'_v(A, y) = P_{\mu(y),v}(A)R(y)$, where*

$$R(y) = \frac{G(y)}{(1 - \epsilon_{\mu(y)}) \int G(y')/(1 - \epsilon_{\mu(y')})dy'},$$

*$G(y)$ is the marginal distribution of $y$ under $Q_v$, which is standard Gaussian.*

*Then $Q'_v(A, y)$ is the contaminated joint distribution of $A, y$ i.e., $Q'_v = (1 - \epsilon)Q_v + \epsilon N_v$ for some noise distribution $N_v$. Under $Q'_v$, it holds that $A|y \sim P_{\mu(y),v}$.*

*Proof.* We proceed to bound $\int G(y')/(1 - \epsilon_{\mu(y')})dy' \leq 1/(1 - \epsilon)$. This shows that $R(y)$ is well-defined, and we will use that to show $Q'_v \geq (1 - \epsilon)Q_v$ (recall that we obscure the line between the distribution and its probability density function).

$$\int G(y')/(1 - \epsilon_{\mu(y')})dy' = \int G(y') \left(1 + \frac{\epsilon_{\mu(y')}}{1 - \epsilon_{\mu(y')}}\right) dy'$$
$$\leq 1 + \int_{|y| \geq c_1} G(y') \left(\frac{\epsilon_{\mu(y')}}{1 - \epsilon_{\mu(y')}}\right) dy' + \int_{|y| < c_1} G(y') \frac{\epsilon}{1 - \epsilon} dy'$$
$$\leq 1 + 2c_1\epsilon + \int G(y')\mu(y')^2 dy'$$
$$\leq 1 + 2c_1\epsilon + \int G(y')c_1^2\epsilon y'^2 dy'$$
$$\leq 1 + 2c_1\epsilon + c_1^2\epsilon$$
$$\leq 1/(1 - \epsilon),$$

where the last step holds by choosing $c_1$ to be sufficiently small.

Since for any $y$ and $A$, $D_{\mu(y)} \geq (1 - \epsilon_{\mu(y)})\mathcal{N}(\mu(y), 1 - c_2\epsilon^2)$ by the construction of $D_\mu$, so $P_{y,v} \geq (1 - \epsilon_{\mu(y)})\mathcal{N}(\mu(y)\text{vec}(vv^\top), I - c_1^2\epsilon^2 \text{vec}(vv^\top)\text{vec}(vv^\top)^\top)$, since $P_{y,v}$ agrees with Gaussian in all directions except $\text{vec}(vv^\top)$. We proved that $R(y) \geq (1 - \epsilon)G(y)/(1 - \epsilon_{\mu(y)})$, so $Q'_v = P_{\mu(y),v}R(y) \geq (1 - \epsilon)G(y)\mathcal{N}(\mu(y)\text{vec}(vv^\top), I - c_1^2\epsilon^2 \text{vec}(vv^\top)\text{vec}(vv^\top)^\top) = (1 - \epsilon)Q_v$. $\square$

Next we show that $Q'_v$ are near orthogonal to each other for distinct $v$, if we view $\chi_S(\cdot, \cdot) - 1$ as an inner product on the space of $d^2 + 1$-dimensional distributions.

**Lemma D.10** (Near Orthogonality of Corrupted Joint Distributions). *Let $S$ be the joint distribution of $A$ and $y$ when they are independent and entries of $A$ are i.i.d. standard Gaussian and $y \sim R$. Then we have*

$$\chi_S(Q'_v, Q'_{v'}) - 1 = e^{O(1/\epsilon^3)}(v^\top v')^4.$$

*Proof.* By the definition of $\chi_S(Q'_v, Q'_{v'})$, we have

$$\chi_S(Q'_v, Q'_{v'}) = \int Q'_v(a, y) Q'_{v'}(a, y) / S(x, y) dx dy$$

$$= \int P_{\mu(y), v}(a) P_{\mu(y), v'}(a) R(y)^2 / (G(a)R(y)) dx dy$$

$$= \int \chi_{\mathcal{N}(0, I_{d^2})}(P_{\mu(y), v}, P_{\mu(y), v'}) R(y) dy$$

$$\leq 1 + O\big((v^\top v')^4\big) \int \chi^2(D_{\mu(y), \epsilon}, \mathcal{N}(0, 1)) R(y) dy,$$

where the last inequality follows from Lemma D.8.

Recall $\mu(y) = c_1 \sqrt{\epsilon} y$. The main technical part is to bound the following quantity:

$$\int \chi^2(D_{\mu(y), \epsilon}, \mathcal{N}(0, 1)) R(y) dy \leq e^{O(1/\epsilon)} + \int_{|y| \geq c_1} e^{O\left(\max\{1/\mu(y)^2, \mu(y)^2\}\right)} R(y) dy$$

$$\leq e^{O(1/\epsilon)} + \int_{|y| \geq c_1} e^{O\left(\max\{1/(c_1^4 \epsilon), \mu(y)^2\}\right)} R(y) dy$$

$$\leq e^{O(1/\epsilon)} + e^{1/(c_1^4 \epsilon)} \int_{|y| \geq c_1} e^{c_1^2 \epsilon y^2} R(y) dy$$

$$\leq e^{O(1/\epsilon)} + e^{1/(c_1^4 \epsilon)} \int_{|y| \geq c_1} e^{c_1^2 \epsilon y^2} (1 - \epsilon) G(y) / (1 - \epsilon_{\mu(y)}) dy$$

$$\leq e^{O(1/\epsilon)} + e^{1/(c_1^4 \epsilon)} \int_{|y| \geq c_1} e^{c_1^2 \epsilon y^2} c_1^2 y^2 G(y) dy$$

$$= e^{O(1/\epsilon)},$$

where the first inequality splits into two cases as in Lemma D.6, and the last inequality follows by choosing $c_1$ to be a sufficiently small constant so that $c_1^2 \epsilon \leq 1/4$ and $\int e^{c_1^2 \epsilon y^2} y^2 G(y) dy$ is bounded above by a constant. $\qquad\square$

*Proof of Theorem 4.2.* The proof follows a standard non-Gaussian component analysis argument. For any $0 < c < 1/2$, there is a set $S$ of at least $2^{d^c}$ unit vectors in $\mathbb{R}^d$ such that for each pair of distinct $v, v' \in S$, $|v^\top v'| = O(d^{c-1/2})$. By Lemma D.10, we have

$$\chi_S(Q'_v, Q'_{v'}) - 1 = \begin{cases} e^{O(1/\epsilon)} O(d^{4c-2}) =: \gamma & \text{if } v \neq v' \\ e^{O(1/\epsilon)} =: \beta & \text{if } v = v' \end{cases}.$$

Therefore, by Lemma 8.8 in [DK23], any SQ algoritm requires at least $2^{d^c} \gamma / \beta = 2^{d^c} / d^{2-4c}$ calls to $\mathsf{STAT}(e^{1/\sqrt{\epsilon}} / d^{1-2c})$ to find $v$ and therefore $u$ within better than $O\big(\epsilon^{1/4}\big)$. $\qquad\square$

We state the auxiliary facts that we used in the above analysis:

**Fact D.11** ([Dia+21, Lemma G.3]). *Let $k \in \mathbb{Z}_+$, distributions $P_i$ and $\lambda_i \geq 0$, for $i \in [k]$ such that $\sum_{i=1}^k \lambda_i = 1$. We have that $\chi^2\left(\sum_{i=1}^k \lambda_i P_i, D\right) = \sum_{i=1}^k \sum_{j=1}^k \lambda_i \lambda_j \chi_D(P_i, P_j)$.*

**Fact D.12** ([DKS19, Lemma F.8]).

$$\chi_{\mathcal{N}(0,1)}\big(\mathcal{N}(\mu_1, \sigma_1^2), \mathcal{N}(\mu_2, \sigma_2^2)\big) = \frac{\exp\left(-\frac{\mu_1^2(\sigma_2^2 - 1) + 2\mu_1\mu_2 + \mu_2^2(\sigma_1^2 - 1)}{2\sigma_1^2(\sigma_2^2 - 1) - 2\sigma_2^2}\right)}{\sqrt{\sigma_1^2 + \sigma_2^2 - \sigma_1^2 \sigma_2^2}}.$$

# E   A Counterexample: Why Simple Algorithms Do Not Work

While Section D formally showed that a broad class of algorithms with sample complexity proportional to the dimension $d$ cannot achieve $O(\epsilon^{1/4})$ error without an exponential amount of computation, we informally discuss a concrete construction of adversarial corruptions where some straightforward

algorithms fail. This discussion is intended only to provide intuition; the SQ lower bound provides more rigorous evidence that improving sample complexity beyond quadratic is impossible. The $O(\cdot)$ notation in this section only displays dependence on the dimension $d$.

Let $(A_i, y_i) \sim \mathcal{G}$ be sampled according to the noiseless Gaussian design in Definition 1.6, i.e. entries of $A_i$ are i.i.d. standard Gaussians and $y_i = \langle M^*, A_i \rangle$.

- The adversary first inspects all $n$ samples and computes $M^*$ up to some small error with out-of-shelf low rank matrix sensing algorithms. In the following discussion, we simply assume the adversary finds $M^*$ exactly.

- The adversary discards a random $\epsilon$-fraction of samples and let $\mathcal{S}$ denote the set of remaining samples. The adversary computes $P = -\frac{1}{n} \sum_{i \in \mathcal{S}} y_i A_i$.

- For $j = 1, \ldots, \epsilon n$, the adversary independently samples $z_j$ uniformly between the spheres of radius $\|M^*\|_F / 2$ and $2 \|M^*\|_F$ and computes $E_j = \frac{P}{\epsilon z_j} + A'_j$, where entries of $A'_j$ are i.i.d. sampled from standard Gaussian.

- The adversary adds $(z_j, E_j)$ as corruptions. Let $\mathcal{B}$ denote the points added by the adversary.

Recall that with the objective function $f_i$ defined in Equation (3) and $B_i = A_i + A_i^T$, we have the gradient and Hessian

$$\nabla f_i(U) = (\langle UU^\top, A_i \rangle - y_i)(A_i + A_i^\top)U$$
$$\nabla^2 f_i(U) = (\langle UU^\top, A_i \rangle - y_i)I_r \otimes B_i + \text{vec}(B_iU)\,\text{vec}(B_iU)^\top.$$

Prior works on first-order robust stochastic optimization [Pra+20; Dia+19] required $O(d)$ samples, because they rely on the robust estimation of the gradients. They would accept 0 as the solution, because when $U = 0$, $\nabla f_i(U) = 0$ for all $i$, i.e., all sample gradients are zero at $U = 0$ whether or not they are corruptions, so gradient-based filtering does not work. On the other hand, $\nabla^2 \bar{f}(0) = -\mathbb{E}[\langle M^*, A_i \rangle I_r \otimes B_i] = -I_r \otimes \mathbb{E}[\langle M^*, A_i \rangle (A_i + A_i^T)] = -2I_r \otimes M^*$, where the last equality follows from Lemma C.2. Hence the error measured by the negative curvature of the solution is $\lambda_{\min}(\nabla^2 \bar{f}(0)) = 2\sigma_1^\star$, which does not even scale with $\epsilon$.

This adversary can also help illustrate why spectral initialization similar to the algorithm discussed in Theorem C.13 cannot succeed with $O(d)$ samples. Spectral methods rely on the observation that with clean samples, $\mathbb{E}_{(A_i, y_i) \sim \mathcal{G}}[y_i A_i] = M^*$. The hope is that the matrix formed by the leading $r$ singular vectors of some robust mean estimation of $\{y_i A_i\}$ is close enough to $M^*$. If so, it can be a good initialization to Algorithm 3.1, which only requires robust estimation of sample gradients and $O(d)$ samples suffice.

However, with $O(d)$ samples, it is unclear how to estimate the mean of $\{y_i A_i\}$. Robust mean estimation algorithms (e.g. Algorithm A.2) discussed in Proposition 2.3 do not work because they require $O(d^2)$ samples. Here we show that the naive filter based on the norm of $\{y_i A_i\}$ does not work, either.

Observe that $P \approx -M^*$ by construction. For each $j$,

$$\|E_j\|_F \le \|A'_j\|_F + \|P\|_F / (z_i \epsilon) \approx \|A'_j\|_F + \|M^*\|_F / (z_i \epsilon) \le \|A'_j\|_F + 2/\epsilon.$$

Since $\|A'_j\|_F$ scales with the dimension $d$ but $\epsilon$ is a fixed constant, as we move to a high-dimensional regime, $\|A'_j\|_F$ dominates and the effects of the outliers diminish if we only look at the norm of data.

Without an effective filter, the mean of $\{y_i A_i\}$ across both clean samples and outliers becomes

$$\frac{1}{n} \sum_{i \in \mathcal{S}} y_i A_i + \frac{1}{n} \sum_{j \in \mathcal{B}} z_j E_j = -P + \frac{1}{n} \sum_{j \in \mathcal{B}} P/\epsilon + A'_j = \frac{1}{n} \sum_{j \in \mathcal{B}} A'_j,$$

which contains no signals — only noise — and is close to 0.

In summary, we constructed an adversary under which robust first-order methods and spectral methods with naive filter fail. Note that this construction relies on the ability of the adversary to corrupt the input matrices $A_i$; if the adversary could only corrupt $y_i$, then the results in [Li+20b; Li+20a] would apply and $\widetilde{O}(d)$ samples would suffice.