# OpenReview forum: "Robust Second-Order Nonconvex Optimization and Its Application to Low Rank Matrix Sensing"
_NeurIPS.cc/2023/Conference — NeurIPS 2023 poster_

### Official Review · Reviewer_d53A · 2023-06-24

**Soundness:** 3 good
**Presentation:** 3 good
**Contribution:** 3 good
**Rating:** 6
**Confidence:** 2

**Summary:**

The authors focus on the problem to find approximate second-order stationary points (SOSPs) in the strong contamination model, where they propose an efficient algorithm with an approximate SOSP as an output. The algorithm is proved to have dimension-independent accuracy guarantees. In particular, the proposed algorithm can solve the low rank matrix sensing problem robustly. They first introduce the formulation of generic nonconvex optimization problem and the low rank matrix sensing and the assumptions for the main theorem where the outputs of the algorithm for the corrupted Stochastic optimization problem can achieve approximate SOSP with high probability. For general cases, the algorithm can obtain approximate SOSPs under strong contamination. At last the authors provide the theoretical guarantees for robust low rank matrix sensing problem.


**Strengths:**

The paper is well written and clear, the approach is well supported by the theoretical analysis. The application on robust low rank matrix sensing problem seems strong and nice.

**Weaknesses:**

- It might be good to include some simulations or real data applications on the robust low rank matrix sensing problem to address its computational efficiency. Or maybe some comparisons to first order method such as  projected gradient descent for robust stochastic optimization problem will be better.

- The authors present theorem showing the results for general robust nonconvex optimization, like Theorem 1.5 and Theorem 3.1. As the authors say below Theorem 1.5, the assumptions appear restrictive and are satisfied for the matrix sensing. My question is whether the general results can hold for other nonconvex optimization problems such as matrix completion, phase retrieval, etc. Or what assumptions might be violated in these cases?

**Questions:**

See weakness.

**Limitations:**

I have not found the part explicit addressing the limitations of the algorithm. Maybe authors can elaborate more on these.

---

> ### Author Rebuttal · Authors · 2023-08-10
>
> About Weakness 1: We addressed simulations and experiments in the global response. To the best of our knowledge, the only first-order method that can robustly find approximate second-order stationary points is [Yin+19]; note that projected gradient descent can only find approximate first-order stationary points, which are not sufficient for a number of nonconvex optimization problems (see Lines 36–57 in Section 1 for the detailed reason). However, the error guarantee in [Yin+19] scales with the dimension, which is very unfavorable and usually uninformative for high-dimensional problems.
>
> About Weakness 2: We addressed why the assumption that the iterates stay in a bounded region $\mathcal{B}$ is fairly general in the global response. For example, none of the mentioned nonconvex optimization problems violate those assumptions of Theorem 1.5. Regarding extensions to other nonconvex problems:
> - For matrix completion, see the global response as to why it may not be interesting in the strong contamination model.
> - For phase retrieval, our techniques should apply and this is an exciting avenue for future work.
>
> About Limitations: As a theory submission to NeurIPS, we stated all assumptions in a precise way. Our algorithms will succeed under those assumptions. As the “Limitations” section is not compulsory in NeurIPS 2023, we did not go into depth to discuss them because all our assumptions are fairly standard.

---

> > ### Comment · Reviewer_d53A · 2023-08-16
> >
> > I would like to thank the authors for their reply. My assessment remains inclined to the positive.

---

### Official Review · Reviewer_jUUi · 2023-07-03

**Soundness:** 3 good
**Presentation:** 4 excellent
**Contribution:** 3 good
**Rating:** 6
**Confidence:** 4

**Summary:**

This paper considers the problem of finding a second-order stationary point in corrupted settings. In particular, it considers the adversarial settings, where a fraction of the observations are arbitrarily corrupted after they are observed. In this setting, under certain assumptions on the cost functions, the authors give an algorithm that yields a second-order stationary point with $n=O(D^2/\epsilon)$ samples. This result is applied to problem of robust matrix sensing.

**Strengths:**

- This work yields the strongest guarantees of finding a second-order stationary point in this setting.
- This is the first work to use both robust mean and robust Hessian approximation to solve the second-order approximation problem.
- The work provides a statistical query lower bounds for rank one matrix sensing that shows that exponentially many queries would be needed to go beyond the $O(d^2)$ bound.

**Weaknesses:**

- One of the assumptions - that the iterates of the algorithms stay in a bounded region $\mathcal B$, is hard to check or guarantee in general.
- The work is a combination of two past works: the robust mean estimation algorithm of DKP20 and an anonymous work Aut23. It is not clear what new theoretical techniques are employed, or if this work is just a combination of the previous.
- The statistical query result also follows the theorems in past works.

**Questions:**

- I wonder when one would want to apply such a method over first-order methods. In particular, this method requires computing estimates of the Hessian, which is quite expensive in high-dimensional settings. Can the authors motivate a bit more why the second-order stationarity is important, for example, in the application they discuss (low-rank matrix sensing)? Experiments on some real data settings could show why one might be interested in second order stationarity.

**Limitations:**

- While an interesting and complete work, I am left unsure of what is new in this work, or if this is just applying already developed techniques to a new setting (finding a second-order stationary point) -- i.e., just combining the results of DKP20 and Aut23.
- No experiments are given showing the practicality of the method, which could help to motivate the usefulness of finding SOSPs.
- This is a limitation of the setting, but many more samples are needed than the information-theoretic threshold.
- It is not clear why it is useful to find second-order stationary points in general.

---

> ### Author Rebuttal · Authors · 2023-08-10
>
> For Weaknesses, we responded to the restrictiveness of the bounded region assumption in the global response. We provided details of our novel theoretical techniques in Section 1.2, starting Line 152. Our main conceptual contribution is to propose a unified framework for designing provably robust learning algorithms by robustly finding second-order stationary points (SOSPs), which is important because for many nonconvex problems, first-order stationarity is not sufficient but second-order stationarity implies global optimality. We then showcase our framework by applying it to solve robust matrix sensing in the strong contamination model with dimension-independent error, even achieving exact recovery when the measurements are noiseless. On a technical level, showing that all iterates stay inside the region using the dissipativity condition is a useful demonstration and technically challenging.
>
> On the lower bound side, although the generic techniques for our SQ lower bound construction exist in the literature, it is highly non-trivial to establish the moment-matching construction required to prove our lower bound. Moreover, it is by no means clear that such a lower bound should hold — there exist efficient algorithms that can estimate the largest eigenvalue (and an associated eigenvector) of a real symmetric positive semidefinite matrix with a sample size proportional to the dimension $d$, and one might think estimating the smallest eigenvector of a symmetric matrix is equally simple. Our SQ lower bound rules out this possibility, because otherwise both gradient directions and negative curvature directions can be estimated with $O(d)$ samples, contradicting the lower bound.
>
> For Questions, Lines 36–57 in Section 1 discussed why we need approximate second-order stationary points. In summary, for a number of nonconvex problems, second-order stationarity implies global optimality, while first-order stationarity is known to be lacking [CG18]. In particular, this statement holds for low-rank matrix sensing problems.
>
> For Limitations about the large sample size, while there may be a gap between the sample size required by our algorithm and the information-theoretic lower bound of the outlier robust low-rank matrix sensing problem, we provided an SQ lower bound that justifies such a gap and shows that our dependence on the matrix dimension $d$ is tight.

---

> > ### Comment · Reviewer_jUUi · 2023-08-18
> >
> > I thank the authors for their response to my questions. I think the lack of practical implementation is still a major weakness, but I am inclined towards a more positive score and will raise it to 6.

---

### Official Review · Reviewer_emkc · 2023-07-05

**Soundness:** 3 good
**Presentation:** 2 fair
**Contribution:** 3 good
**Rating:** 5
**Confidence:** 3

**Summary:**

In this work, the authors proposed a new algorithm to find approximate second-order stationary points for stochastic optimization problems under the strong contamination model. The general algorithm is applied to the robust matrix sensing problem and the convergence results are proved for the robust matrix sensing problem.

**Strengths:**

The results in this work are novel and should be interesting to audiences in optimization and machine learning fields.

**Weaknesses:**

It is unclear how the results of this work differ with those in literature; see my comment (10) in the next section. The presentation of the results and the sketch of proofs can be improved. Currently, many important technical details are omitted in the main manuscript. For example, Algorithms A.1-2 and the construction of distributions for the SQ lower bound.

**Questions:**

(1) Line 48: "newdiscussed" is a typo.

(2) Line 69: it would be better to (briefly) discuss the counterexample in the appendix.

(3) Line 85: it seems that both "second-order stationary point" and the abbreviation "SOSP" are used throughout the paper. It would be better to be consistent in using the abbreviation.

(4) Line 93: I think B_D_g and B_D_H should be the bound on the norm of the gradient and the Hessian matrix, respectively?

(5) Theorem 1.5: maybe the authors can briefly discuss the reason why the sample complexity is inversely proportional to the corruption rate \epsilon?

(6) Line 148: it may be better to use D to denote the dimension.

(7) Line 157: I wonder if the region is the same as the region B in Assumption 1.4? In addition, is the information about region B provided to the algorithm as an input parameter? It will be helpful if the authors can clarify it in the paper.

(8) It seems that the statements on Lines 163 and 166 are the same.

(9) Line 169: it would be better if the authors can be more specific on the circular dependence. Does it refer to the rotational invariance of the objective function?

(10) Line 176: I think the results in the following paper also concerns the noiseless and Gaussian measurement case. It will be better if the results in the following paper can be discussed and compared.

Li, Xiao, Zhihui Zhu, Anthony Man-Cho So, and Rene Vidal. "Nonconvex robust low-rank matrix recovery." SIAM Journal on Optimization 30, no. 1 (2020): 660-686.

(11) In Equation (1), the computational complexity is proportional to \epsilon_g^2. This is a little counter-intuitive, since the upper bound of the running time is not changed is we shrink \epsilon_g and \epsilon_H together (at different rates). It would be better if the authors can include an explanation to this relation.

(12) Line 231: "exists"

(13) Theorem 3.5: It would be better to mention the condition \epsilon = O(1/k^2/r^2).

(14) Line 320: "Theorem 3.1 applies"

(15) Line 367: it would be better to explain why the results are more significant or important for the rank-1 case.

(16) Line 378: it will be helpful if the authors can briefly explain how to generate the SQ oracle.

**Limitations:**

See my comments in the previous section.

---

> ### Author Rebuttal · Authors · 2023-08-10
>
> We thank the reviewer for pointing out the typos in Questions (1)(6)(12)(14) and suggestions for our presentation Questions (3)(13). Question (15) was addressed in the global response. As for the weakness about our presentation, we provided a sketch of the proof in the main body and deferred algorithms introduced in other papers and technical details to the appendix due to page limitations.
>
> About Question (10) and the main weakness about our novelty: Thank you for pointing out this improvement over [Li+20] and we will add it to the references. This paper, as well as the cited [Li+20], discussed the setting where outliers only exist in the measurements $y_i$. Our paper, in contrast, considers the more challenging scenario where both sensing matrices $A_i$ and measurements $y_i$ can be corrupted. Since the sensing matrices are high-dimensional objects, allowing their corruptions results in a much more difficult problem: while both [Li+20] and [Li+20a] can solve their problem with $\widetilde O(d r)$ samples, our setup requires $\widetilde \Omega (d^2)$ samples as evidenced by our SQ lower bound.
>
> [Li+20a] Xiao Li, Zhihui Zhu, Anthony Man-Cho So, and Rene Vidal. "Nonconvex robust low-rank matrix recovery." SIAM Journal on Optimization 30, no. 1 (2020): 660-686.
>
> About Question (5): See the global response for the derivation of the sample complexity. Our algorithm uses robust mean estimation discussed in the global response as a subroutine, which is why we have the same dependence on $\epsilon$.
>
> About Question (7): Yes, the region in Theorem 1.5 is the same as in Assumption 1.4. Our algorithm does not explicitly check whether the iterates move outside the region. This is a fairly standard assumption in robust nonconvex optimization (see the global response where we argue this assumption is not restrictive). In the application to low-rank matrix sensing, we proved that this assumption is indeed satisfied. Note that information about region $\mathcal{B}$, though not explicitly required and checked, might be implicitly passed to the algorithm because it can be related to the Lipschitz constant of the gradient or Hessian of the objective function.
>
> About Questions (8) & (9): The circular dependency in Line 169 refers to the following dependencies of three quantities: the gradient inexactness estimate depends on the covariance bound of gradients (which depends on the covariance bound of the gradients, see Proposition 2.3), the distance between current solution and global optima (which depends on the gradient inexactness, see Equation (32)), and the covariance bound of the gradients (which depends on the distance to global optima, see Equation (17) when we computed the covariance of the gradients). We intended to use this paragraph to sketch and help readers understand the proof of Theorem 3.8 and its difficulties. The repetition of similar wordings in Lines 163-166 happens when we are trying to sketch how we overcame such circular dependence.
>
> About Question (2): There is a counterexample from our low-rank matrix sensing application where the adversary can trick the algorithm into believing that a saddle point with dimension-dependent negative curvature appears to be a local minimum solution. We will include this counterexample in the appendix.
>
> About Question (4): When we discuss the general robust nonconvex optimization result, we consider the gradient to be a vector and Hessian to be a matrix. When we apply this result to robust low-rank matrix sensing, we vectorize the matrix variable $U$ when we invoke Algorithm A.1 to fit into the general framework.
>
> About Question (11): Equation (1) consists of two terms. The first term is the expected computational complexity that scales with $\max(\epsilon_g^{-2}, \epsilon_H^{-3})$, which is standard for vanilla gradient descent algorithms with negative curvature detection. The second term is the correction when we extend the expected computational complexity to the high probability bound, which involves some more complicated interplay between the gradient step ($\epsilon_g$ dependence) and negative curvature steps ($\epsilon_H$ dependence). If we shrink $\epsilon_g$ and $\epsilon_H$ together, the first term will increase.
>
> About Question (16): A statistical query with accuracy \tau can be implemented with error probability $\delta$ by taking $O(\log(1/\delta)/\tau^2)$ samples and evaluating the empirical mean of the query function $q(\cdot)$ evaluated at those samples. It is usually possible to reuse samples across different queries. See the paragraph “SQ Algorithms versus Traditional Algorithms” in [DK23, Chapter 8.2.1] for a more detailed discussion.

---

> > ### Comment · Reviewer_emkc · 2023-08-20
> >
> > I would like to thank the authors for their detailed response! I will remain netural and lean towards accept. So I will keep my score.

---

> ### Comment · Area_Chair_hPbb · 2023-08-20
> **Response to Authors' Rebuttal?**
>
> Dear Reviewer emkc,
>
> Thanks for your hard work in reviewing this paper. As the author-reviewer discussion period will end on Monday, August 21, could you kindly take a look at the authors' responses to your comments and indicate whether you are satisfied with them? Your timely feedback not only will contribute greatly to the decision process but will also be greatly appreciated by the authors and program team.
>
> Best,
> Your AC

---

### Official Review · Reviewer_kWga · 2023-07-09

**Soundness:** 3 good
**Presentation:** 3 good
**Contribution:** 3 good
**Rating:** 6
**Confidence:** 3

**Summary:**

This paper studies the problem of finding approximate second-order stationary point when a constant fraction of datapoints are corrupted by outliers. It proposes an algorithm with provable guarantees which matches the statistical query lower bound established in the paper. The general result is applied to study low-rank matrix sensing with outliers.

**Strengths:**

This paper first proposes a general result in finding approximate SOSP and then applies it to study the widely applicable problem of low-rank matrix sensing. The newly proposed algorithm is proved to be able to find an approximate SOSP in polynomial time even when a constant proportion of samples are corrupted. It also provides a lower bound on the sample complexity for rank-one matrix sensing which matches the required sample size of the algorithm, confirming the efficacy of the algorithm.

**Weaknesses:**

1. The results obtained in Theorem 1.7 and 3.3 suggest that increasing the sample size $n$ (while fixing the noise level $\sigma$ and outlier fraction $\epsilon$) does not enhance the algorithm's performance, as the estimation error does not depend on the sample size. This observation seems counterintuitive and calls for additional clarification in order to better understand this phenomenon.

2. In Section 4, the paper presents a lower bound for low-rank matrix sensing specifically for the case where the rank is one (i.e., $r=1$). The statement "Our main result in this section is a near-optimal SQ lower bound for robust low-rank matrix sensing that applies even for rank $r=1$" suggests that the rank-one scenario poses additional challenges than the general low-rank case, which doesn't sound reasonable. Further clarification is needed to better understand this.

**Questions:**

1. This paper considers symmetric case. What if the matrix $M^{\star}$ is asymmetric?

2. A closely related problem is matrix completion. Is it possible to extend the analysis to matrix completion?

3. It is assumed that the algorithm requires knowledge of a multiplicative upper bound $\Gamma$ of $\sigma_1^{\star}$. How can this parameter be estimated in practice?

**Limitations:**

This paper does not have potential negative social impact.

---

> ### Author Rebuttal · Authors · 2023-08-10
>
> Weaknesses 1 & 2 and Question 2 were addressed in the global response.
>
> For Question 1, we intended to present our robust low-rank symmetric matrix sensing algorithm as a simple application of our general robust nonconvex optimization framework. The techniques are generalizable to asymmetric ground truth matrix $M^*$: [GJZ17, Section 5] discussed how to reduce asymmetric $M^*$ to the symmetric case, i.e. when $M = U V^\top$ for some tall matrices $U$ and $V$, we have $[UU^\top, M; M^\top, VV^\top] = [U; V] [U; V]^\top$ (which is symmetric) and one can add an additional regularizer in the objective function so that $UU^\top \approx VV^\top$. We leave the details to future work.
>
> For Questions 3 & 4, knowing an upper bound of the norm of the ground-truth matrix is a standard assumption in matrix sensing, even for non-robust settings. See, e.g., [GJZ17] and [Jin+17]. Its estimation is an interesting question but beyond the scope of our paper. One potential technique is to double the upper bound until a solution is found; the estimation procedure can certify a solution by robustly estimating the value of the objective function and checking whether the error bound in Theorem 1.7 is satisfied. We will clarify this.

---

> > ### Comment · Reviewer_kWga · 2023-08-20
> >
> > Thank you for the clarification! I will increase the score accordingly.

---

### Author Rebuttal · Authors · 2023-08-10

We thank the reviewers for their careful consideration of our work and the positive feedback. Below we address some common concerns raised by the reviewers. We hope that the provided clarifications will help clear possible misunderstandings and elevate the reviewer’s assessment of our contributions.

**On the restrictiveness of the assumption that all iterates stay inside a bounded region $\mathcal{B}$ in Theorem 1.5**: As demonstrated in our low-rank matrix sensing example, this assumption is easy to satisfy if the objective function satisfies a dissipativity condition. The dissipativity condition says that the gradient aligns with the iterate when its norm exceeds some threshold; this threshold determines the radius of $\mathcal{B}$, which is allowed to depend polynomially on the dimension (see Line 97). Dissipativity is a fairly general phenomenon [Hal10]. See also [RRT17, Section 4] for a discussion of how adding a $\ell_2$-regularization term enables a Lipschitz function to satisfy the dissipativity condition.

[RRT17] Maxim Raginsky, Alexander Rakhlin, and Matus Telgarsky. "Non-convex learning via stochastic gradient Langevin dynamics: a nonasymptotic analysis." In Conference on Learning Theory, pp. 1674-1703. PMLR, 2017.

**On the choice of rank r = 1 in the construction of SQ lower bound**: The case where the rank $r = 1$ is in fact the easiest parameter regime, hence leading to a stronger lower bound. Recall that the sample complexity of our algorithm is $\widetilde O(d^2 r^2)$ as in Theorems 3.2 and 3.3, and the main point of our SQ lower bound shows that the $d^2$ factor is necessary *even if* $r = 1$.

**On the sample size in Theorems 1.7 and 3.3**: The error rate under the strong contamination model (Definition 1.1) usually takes the form of $\widetilde O(f(\epsilon) + g(d/n))$, where $f$ and $g$ are some nondecreasing functions, $n$ is the number of samples, and $d$ is the dimension of the samples. Even with infinite samples, the contribution of the error rate from $\widetilde O(f(\epsilon))$ does not go to 0. For example, any robust estimator for the mean of a $d$-dimensional identity covariance Gaussian must incur $\ell_2$-error $\Omega{\epsilon}$ in the strong contamination model [DK23, Chapter 1.2]. We choose the sample size $n$ so that the contribution from $\widetilde O(g(d/n))$ is comparable to $\widetilde O(f(\epsilon)$, as is the standard practice in algorithmic robust statistics. For the robust mean estimation subroutine used in our paper, Proposition 1.5 in [DKP20] showed that $\epsilon$-corruption of $d$-dimensional samples from a distribution with covariance bounded by  $\sigma^2 Identity$ gives error rate of $\widetilde O(\sqrt{d\sigma/n} + \sqrt{\sigma \epsilon})$. Matching the contribution from the first term with the second term gives the sample size $n = \widetilde O(d / \epsilon)$.

**On the robust matrix completion problem**: The general robust nonconvex optimization framework and the techniques developed in this paper extend to the robust matrix completion problem (where both values and locations of a fraction of observed entries are corrupted). However, under our problem setup and with our current results, matrix completion where the target matrix has a total of $d^2$ entries requires a sample size bound of $O(d^2)$ (which is potentially optimal if our lower bound for matrix sensing extends to matrix completion). This does not seem to be a particularly interesting result, because this sample complexity allows the algorithm to inspect all entries multiple times. For some other contamination models (different from Definition 1.1), only $\widetilde O(d \operatorname{poly}(r))$ samples are needed (see, e.g., [CG18] and [CGJ17]). Those would be more reasonable sample sizes but their settings are drastically different from ours.

[CGJ17] Yeshwanth Cherapanamjeri, Kartik Gupta, and Prateek Jain. "Nearly optimal robust matrix completion." In International Conference on Machine Learning, pp. 797-805. PMLR, 2017.

**On lacking simulations and experiments**: Our work is a learning and optimization theory submission that is well within the scope of NeurIPS (based on the Call for Papers). The sample/computational efficiency and error guarantees of our methods are analyzed in a precise way. We believe that our technical results are theoretically interesting and request that our work be judged based on its merits. That said, we acknowledge that experimental evaluation is important and a fruitful direction for future work.

---

### Decision · Program_Chairs · 2023-09-21

**Decision:**

Accept (poster)

**Comment:**

The reviewers generally find that the paper has good theoretical contributions on the problem of finding approximate second-order stationary points of outlier-robust stochastic optimization problems. However, they also point out the insufficiency of the numerical evaluation. When preparing the camera-ready version, please incorporate the reviewers' suggestions and address their comments carefully.